# Why flying insects gather at artificial light

Samuel T. Fabian [1,5] ✉, Yash Sondhi [2,3,5] ✉, Pablo E. Allen[4], Jamie C. Theobald[2,6] & Huai-Ti Lin [1,6]

Explanations of why nocturnal insects fly erratically around fires and lamps have included theories of "lunar navigation" and "escape to the light". However, without three-dimensional flight data to test them rigorously, the cause for this odd behaviour has remained unsolved. We employed high-resolution motion capture in the laboratory and stereo-videography in the field to reconstruct the 3D kinematics of insect flights around artificial lights. Contrary to the expectation of attraction, insects do not steer directly toward the light. Instead, insects turn their dorsum toward the light, generating flight bouts perpendicular to the source. Under natural sky light, tilting the dorsum towards the brightest visual hemisphere helps maintain proper flight attitude and control. Near artificial sources, however, this highly conserved dorsal-light-response can produce continuous steering around the light and trap an insect. Our guidance model demonstrates that this dorsal tilting is sufficient to create the seemingly erratic flight paths of insects near lights and is the most plausible model for why flying insects gather at artificial lights.

The interaction between flying insects and artificial light, is such a common occurrence that it has inspired the saying "drawn like a moth to a flame"[1]. Artificial light is an ancient method to trap insects, with the earliest written records dating back to the Roman Empire around 1 AD[2,3]. Efforts to improve light trap efficiency have generated many observations about nocturnal phototaxis, including phenomenological data on the effects of wavelength, the moon, sky brightness, and weather[4,5]. Consequently, several qualitative models of how insects gather at light have been proposed[6]. Some of the most popular theories are: (1) Insects are drawn to light through an escape mechanism, directing their flight toward it as they might aim for a gap in the foliage[7]. (2) Insects use the moon as a celestial compass cue to navigate, and mistakenly use artificial light sources instead[8]. (3) Thermal radiation from light sources is attractive to flying insects[9]. (4) The sensitive night-adapted eyes of insects are blinded by artificial lights, causing them to fly erratically or crash, and trapping them near light sources[10,11]. Understanding how insects interact with artificial light is particularly important amid modern increases in light pollution that are a growing contributor to insect declines[12,13].

Compared to the abundance of hypotheses, the kinematic data required to test their predictions are exceedingly rare[11,14]. The thermal radiation model has been conclusively found to be flawed[15], while other models continue to be proposed today[16,17]. Why has a conclusive answer evaded us? In part, because 3D tracking of small flying objects in low light is technically challenging, and necessary tools did not exist[18]. That did not stop researchers from attempting innovative experiments, such as attaching moths to polystyrene boats[11]. However, in-flight 3D flight trajectory and orientation measurements have remained difficult[19,20]. We leveraged advances in camera hardware and tracking software to consider the sensory requirements for insect flight control, and how artificial light may disrupt them.

Flying animals need a reliable way to determine their orientation with respect to the external world, especially with reference to the direction of gravity. Throughout the long evolutionary history of insect flight, the brightest part of the visual field has been the sky, and thus it is a robust indicator of which way is up. This is true even at night, especially at short wavelengths (<450 nm)[21]. Most flying insects display some form of the dorsal-light-response (DLR), a behaviour that keeps their dorsal (top) side to the brightest visual region[22–26]. This has been

[1]Department of Bioengineering, Imperial College London, London SW7 2AZ, UK. [2]Institute for Environment, Department of Biology, Florida International University, Miami, FL 33174, USA. [3]McGuire Center for Lepidoptera and Biodiversity, Florida Museum of Natural History, University of Florida, Gainesville, FL 32611, USA. [4]Council on International Educational Exchange, Monteverde Apto 43-5655, Costa Rica. [5]These authors contributed equally: Samuel T. Fabian, Yash Sondhi. [6]These authors jointly supervised this work: Jamie Theobald, Huai-Ti Lin. ✉e-mail: s.fabian@imperial.ac.uk; yashsondhi@gmail.com

demonstrated in tethered insects but the behavioural phenomenon's effects are difficult to test in free flight[22,23,26–28]. We considered that the presence of an artificial point light source could: (1) reduce or remove the accuracy of the dorsal-light response and mislead insects to tilt their dorsum away from the sky; (2) misdirect lift-generation and disrupt flight-stability; (3) inhibit coherent heading control[26,29]. In contrast, diffused artificial light in the same plane as the night sky should restore appropriate dorsal-light-response, allowing insects to fly normally.

Insects have other possible means of correcting their aerial attitude (orientation with respect to gravity). The largest flying insects, such as dragonflies and butterflies, can leverage passive stability to help stay upright[30,31]. However, the small size of most insects means they travel with a lower ratio of inertial to viscous forces (Reynolds number) compared with larger fliers[32]. Consequently, smaller insects, such as flies, cannot glide or use passive stability, yet must still maintain an appropriate flight attitude and rapidly correct for undesired rotations[33]. Multiple visual and mechanosensory mechanisms contribute to the measurement and correction of undesired rotations, but most measure rotational rate rather than absolute attitude[26,28,32,34]. In environments without artificial light, the brightest portion of the visual environment offers a reliable cue to an insect's current attitude.

To understand the effects of artificial light on insect flight at night, we captured high-resolution flight trajectories across 10 different orders of insects in the presence of artificial lights. This dataset was used to evaluate common models of nocturnal light entrapment, and to establish a model based on the subversion of the DLR. We used point sources, and diffuse illumination in different orientations and collected flight data in broadly two categories: (1) High frame-rate stereo recordings of the flight-paths of wild insects near an artificial light source in field conditions; (2) Captive flight experiments with free-flight body orientations measured with high-resolution motion capture. Our field experiments with light manipulation qualitatively showed strong dorsal tilting behaviour. The motion capture data allowed us to quantitatively probe the aerial manoeuvres of the insects in free flight around light sources. Extensive analyses on 3D flight trajectories helped evaluate competing models. Finally, we reproduced the flight behaviour of light-entrapped insects by simulating the dorsal tilting control objective, demonstrating that a simple behavioural response could underlie the light entrapment phenomenon.

## Results
### Artificial point light source induces abnormal flight behaviour in insects
Across 477 stereo-videographic field recordings (median duration 1.7 s IQR 1.9 s) (Supplementary Fig. 1 & Supplementary Data 1, 2), we identified three visually evident behavioural motifs (Table 1) observed in 10 orders of insect flying near artificial light (Table 2). *Orbiting* could be identified by the relatively stable circular flightpath around the light with sustained speed (Fig. 1a & Supplementary Movie 1). The insect appeared to maintain a stable banked attitude with the body tilted laterally (banked or rolled) towards the light source. Orbiting was prevalent at low wind condition (<1 m/s), with insects dispersing if a

gust of wind arose. *Stalling* was characterised by a steep climb as the insect faced away from the light source (Fig. 1b & Supplementary Movie 2), losing speed until the insect ceased to make progress. *Inversion* of the insect's attitude (either through roll or pitch) occurred when the insect flew directly over a light source (Fig. 1c & Supplementary Movie 3), resulting in a steep dive to the ground. We observed these motifs with insects flying around the light source under all conditions, but rarely (<2%) in videos of insects in the dark, hence our characterising them as abnormal (Table 1). Once below the light, insects frequently righted themselves, only to climb above the light and invert once more. During these flights, the insects consistently directed their dorsal axis towards the light source, even if this prevented sustained flight and led to a crash.

### Motion capture quantifies dorsal tilting toward light
To quantitatively understand the behaviour observed in our field recordings, we used insect-scale motion capture to record flying insects in a controlled behavioural arena (median track duration 1.7 s IQR 4.3 s) (Supplementary Fig. 2 & Supplementary Data 3). Our motion capture used infrared light to track a custom marker frame (<5% of insect bodyweight) mounted onto the thorax of insect subjects, without disrupting their vision. Three markers arranged in an L-shape allowed us to measure the rotations and translations of the frame in space[35], and thus the insect's dorsal axis. For this experiment, we tuned the system for recording volume with acceptable accuracy (marker residuals <0.24 mm, or <7° absolute orientation error for the smallest insect).

We flew different insect species within a two-metre diameter cylindrical tent around 3 different light sources: a UV LED bulb (395 nm), a UV-Blue Actinic tube (spectral peak 370 nm), and a cool-white LED bulb, with no other light source salient to the insect. To test diurnal species not generally associated with light-entrapment, we used the Common Darter (*Sympetrum striolatum*) (*n* = 12) and Migrant Hawker (*Aeshna mixta*) (*n* = 2) dragonflies. For nocturnal species, we used Yellow Underwing Moths (*Noctua pronuba* & *Noctua fimbriata*) (*n* = 8 and *n* = 2 respectively) and Lorquin's Atlas Moth (*Attacus lorquinii*) (*n* = 3). Across these four species we recorded 538 continuous flight trajectories (Supplementary Data 4, Table 3).

When flying around a point light source, flights were visibly disturbed as described by the motifs observed in the field (Supplementary Movie 4). Flight trajectories viewed from above (Fig. 2a) show orbiting around the light, with few direct flights toward the light. We projected the velocity vectors of the 4 species onto the ground plane and compared them to the instantaneous direction of the light (Fig. 2b). In all 4 cases, the velocity vector strongly concentrated orthogonally from the direction of the light source, refuting the idea of flying directly toward the light. We used the Rayleigh z-test to test confirm the clustering of the velocity vectors was significant for all species (*S. striolatum*, z = 1356.10, $p < 0.001$, $n = 7032$ sub-sampled trajectory points; *A. mixta*, z = 102.09, $p < 0.001$, $n = 1085$; *Noctua sp.*, z = 624.90, $p < 0.001$, $n = 1759$, *A. lorquinii*, z = 359.30, $p < 0.001$, $n = 1403$). In the flight arena, Common Darter dragonflies do not exhibit such orbiting pattern under broad spectrum diffused canopy

**Table 1 | Summary of sample sizes of behavioural motifs observed (Total = 477 videos)**

| Treatment | No. of videos | Orbit | Stall | Invert | Total motif count | 0 behaviours (%) | 1 behaviour (%) | 3 behaviours (%) |
|---|---|---|---|---|---|---|---|---|
| No light | 41 | 0 | 3 | 0 | 3 | 92.68 | 7.32 | 0 |
| Sheet down | 46 | 0 | 1 | 42 | 43 | 8.7 | 89.13 | 0 |
| UV bulb down | 125 | 96 | 82 | 83 | 261 | 1.6 | 25.6 | 37.6 |
| UV bulb up | 83 | 16 | 38 | 44 | 98 | 16.87 | 50.6 | 2.41 |
| UV tube | 127 | 87 | 72 | 64 | 223 | 6.3 | 30.71 | 18.9 |
| White sheet above | 55 | 1 | 7 | 2 | 10 | 85.45 | 12.73 | 1.82 |

 

light (z = 0.43, p = 0.36, n = 269) or in pitch-dark (z = 0.43, p = 0.65, n = 304). This demonstrates that *Orbiting* was caused by the UV light source, not the enclosure (Fig. 2).

The marker frame data revealed that the insects strongly tilted their backs towards the light source (Fig. 3). Examining the dorsal axis projected onto the ground-plane showed the body tilt strongly matched the direction of the light with a 1:1 ratio in all four species (Fig. 3a). Insects were tilting their dorsal axes directly toward the light as they flew around it. We took the dot-product between the normalized projected dorsal axes and the light source direction as an index: ranging from −1 (away from the light source) and 1 (toward the light source). Our index values for insects flying around a light source were 0.84 (n = 9904 frames) for *S. striolatum*, 0.79 (n = 1416) for *A. mixta*, 0.82 (n = 1563) for *Noctua sp.*, and 0.82 (n = 1357) for *A. lorquinii*, indicating strong dorsal tilting towards the light in each species. In contrast, with the light off, *S. striolatum* had a tilting direction index of 0.17 (n = 713), indicating weak dorsal tilting consistency in darkness.

We further explored the light-disturbed flight attitude distribution of the four species by plotting their bank and pitch orientation composition (Fig. 3b). To compare this to the undisturbed flights, we allowed dragonflies to fly under bright, broad-spectrum lamps

### Table 2 | Summary of behavioural motifs observed around light sources, separated by insect order (Total = 448 videos)

| Order | Orbit | Stall | Invert | No. Of videos |
|---|---|---|---|---|
| Blattodea | 1 | 0 | 0 | 1 |
| Coleoptera | 7 | 4 | 2 | 8 |
| Diptera | 9 | 4 | 4 | 13 |
| Ephemeroptera | 1 | 0 | 0 | 1 |
| Hemiptera | 3 | 2 | 3 | 4 |
| Hymenoptera | 9 | 9 | 7 | 11 |
| Lepidoptera | 158 | 172 | 222 | 387 |
| Mantodea | 0 | 2 | 1 | 2 |
| Neuroptera | 1 | 1 | 0 | 1 |
| Trichoptera | 3 | 4 | 2 | 4 |
| Unknown | 9 | 9 | 7 | 16 |

illuminating the arena from the ceiling. For the two moth species, we used a single actinic tube to produce a diffuse UV-Blue ceiling, while keeping overall light-levels low. Under control conditions, all four species showed typical cruising level flight distribution with body bank angle below 30° (bank angle medians ± interquartile range: 11.9 ± 14.8° for *S. striolatum*, 13.9° ± 22.7° for *Noctua sp.*, 20.2 ± 21.4° for *A. mixta*, and 8.7° ± 11.2° for *A. lorquinii*), with most variation in pitch as required by routine manoeuvres such as turning, climbing, or descending. The bank-pitch distributions of all species near a point light source differed considerably from their controls. *S. striolatum* and *Noctua sp.* showed strong and high bank near point sources. The larger *A. mixta* and *A. lorquinii* showed less consistent body attitude but still shifted their bank-pitch distribution to higher values when near a point light source. All species showed some degree of higher bank when near a point light source (bank angle medians ± interquartile range: 43.8 ± 39.2° for *S. striolatum*, 48.0° ± 30.7° for *Noctua sp.*, 29.3 ± 30.8° for *A. mixta*, and 30.7° ± 31.8° for *A. lorquinii*) (Wilcoxon rank sum; *S. striolatum*, z = 19.91, p < 0.001, *Noctua sp.*, z = 11.18, p < 0.001, *A. mixta*, z = 4.32, p < 0.001, *A. lorquinii*, z = 16.85.91, p < 0.001). This data suggests that a point light source significantly alters attitude control, as the insects attempt to align their dorsal axis toward the light.

### Sky-like artificial light restores normal flight
An established method for light-trapping insects involves shining a bright light onto a white sheet[36]. In the field, we filmed a shrouded UV light source facing downward (the bulb concealed above) onto a white fabric sheet spread across the ground. In these recordings, we observed insects inverting, and tumbling in the air before crashing into the ground (Fig. 4a). If this trapping effect is mediated by the DLR, we expect insects not to be trapped by otherwise similar light sources that match naturalistic cues. When we used the same shrouded UV bulb to shine upward onto a white sheet stretched above, it created a corridor in which UV-Blue light reflected down as a diffuse canopy similar to the sky. In this arrangement, insects did not fly upward toward the bulb, or cluster around the light, but rather flew various paths under the light through the canopied corridor (Fig. 4b), supporting the notion that crashing behaviour is a consequence of a mismatch between the insects' sense of upward and the true direction of gravity (Fig. 4c).

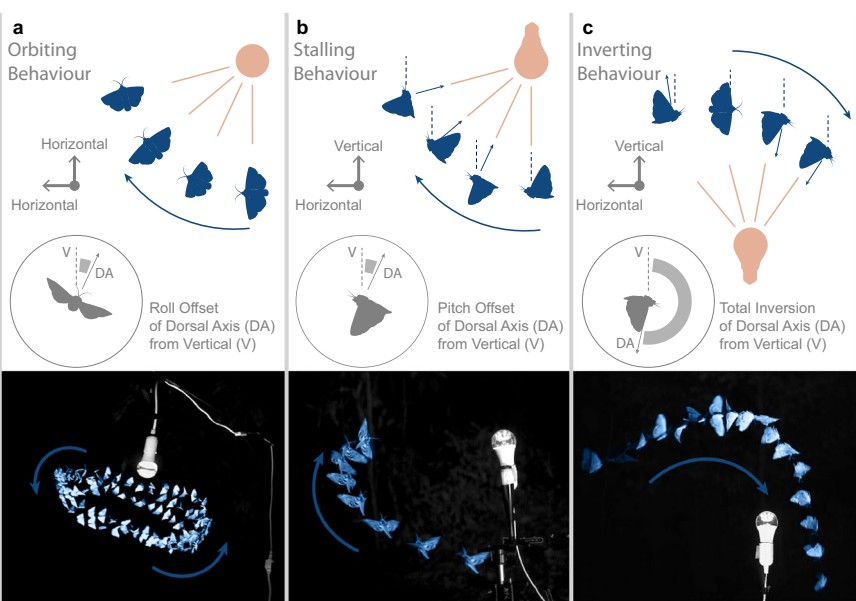

**Fig. 1 | Insects flying around a light source in the field display 3 common behavioural motifs not seen in normal flight.** The unusual flight motifs were: **a** Orbiting, **b** Stalling, and **c** Inverting. (*Above*) Diagrammatic representations of the three behavioural motifs. (*Below*) Overlaid flight trajectories of insects performing these characteristic patterns around UV light sources. Overlaid frames are separated by aesthetically chosen fixed intervals of 52 ms (left), 20 ms (middle), and 24 ms (right) for visualization.

To test whether smaller insects may be more resilient to the manipulation of the dorsal light response, we caught a mixture of small insects (body size 2 cm or below, from 13 families of 6 orders see

### Table 3 | Summary of sample sizes (number of recorded tracks) for the motion capture recordings of different lighting conditions (Total = 599 tracks)

| Species | Actinic tube | Point bulb (downward) | Point bulb (upward) | Control | Total dark |
|---|---|---|---|---|---|
| *Sympetrum striolatum* | 57 | 53 | 61 | 26 | 27 |
| *Aeshna mixta* | 20 | 5 | 36 | 11 | 0 |
| *Noctua sp.* | 90 | 33 | 0 | 20 | 0 |
| *Attacus lorquinii* | 19 | 44 | 25 | 11 | 0 |
| *Daphnis nerii* | 9 | 16 | 36 | 0 | 0 |

Table 4 for taxonomic composition). We placed subjects in a clear Perspex cuboid tank (20 cm on each side) for high-speed filming. With diffuse UV light (~400 nm) from above, all tested small insects flew upward towards the ceiling of the enclosure in a rapid but stable manner, resembling normal escape flight. However, with UV light from below, none of the tested taxa except *Drosophila sp.* (discussed in a later section) were able to maintain flight, tilting and inverting soon after take-off and crashing into the floor (Fig. 4d, Supplementary Movie 5). These results indicate smaller insects also heavily rely on the direction of light to determine the upward direction in flight, and that specific sensory organs such as dipteran halteres do not compensate for inaccurate estimation of verticality. All Diptera were also tested with cool-white LED bulbs above and below their tank. No Diptera exhibit the tipping and crashing behaviour over the white source, suggesting the effect is specific to short wavelengths of light in Diptera tested.

Our qualitative observation that normal flight occurs under a diffuse canopy (Fig. 4b) can be confirmed by some quantitative

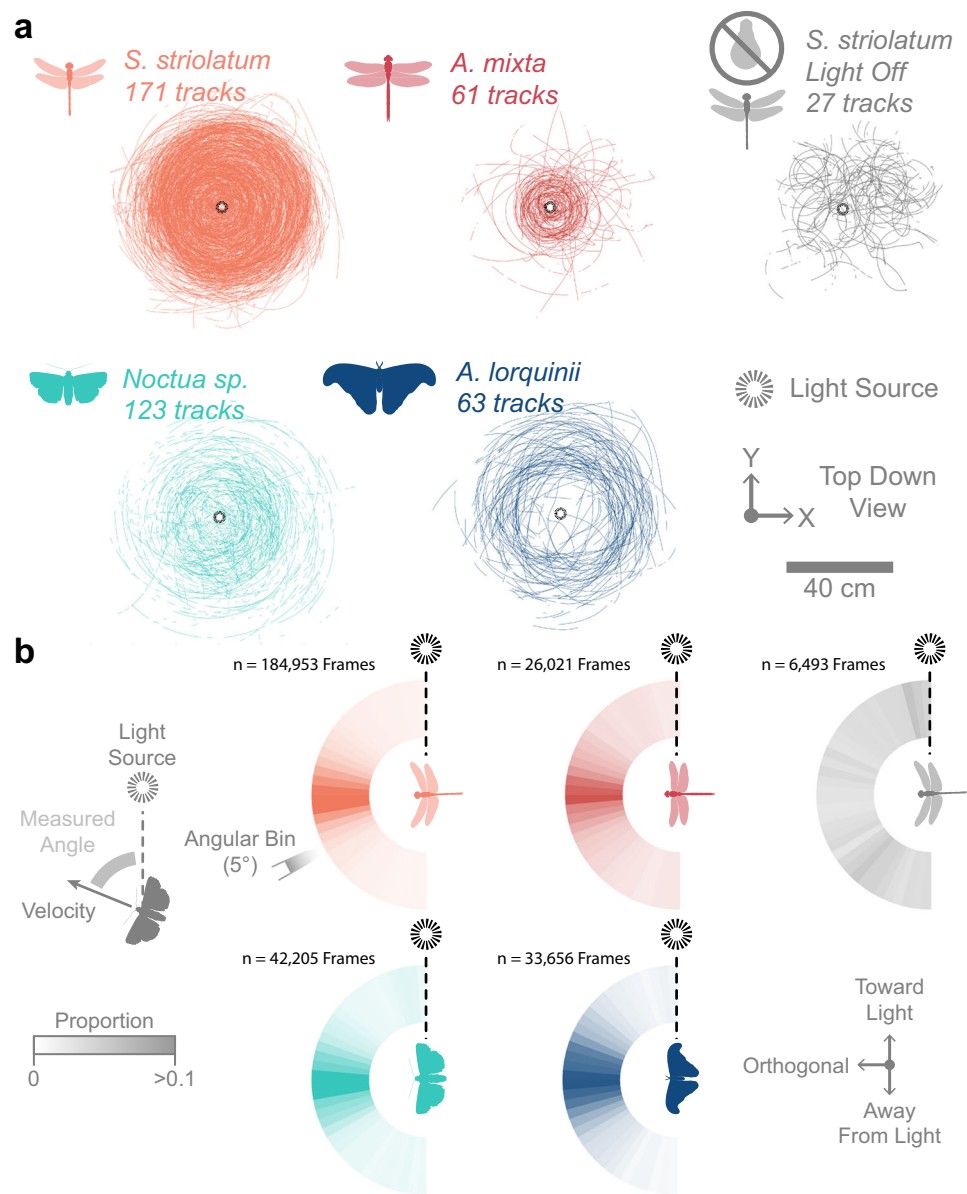

**Fig. 2 | Insects within a controlled environment did not head towards the light source, but predominantly orbited it. a** Top-down plotted flight tracks for each of the 4 main study species with a central downward-facing bulb or vertical tube light source (*left four panels*), and *Sympetrum striolatum* in total darkness (*right*).

**b** The horizontal orientation of insect velocity relative to the light source is given by a radial histogram in which count proportion is colour-coded within each 5° bin. Source data are provided as a Source Data file.

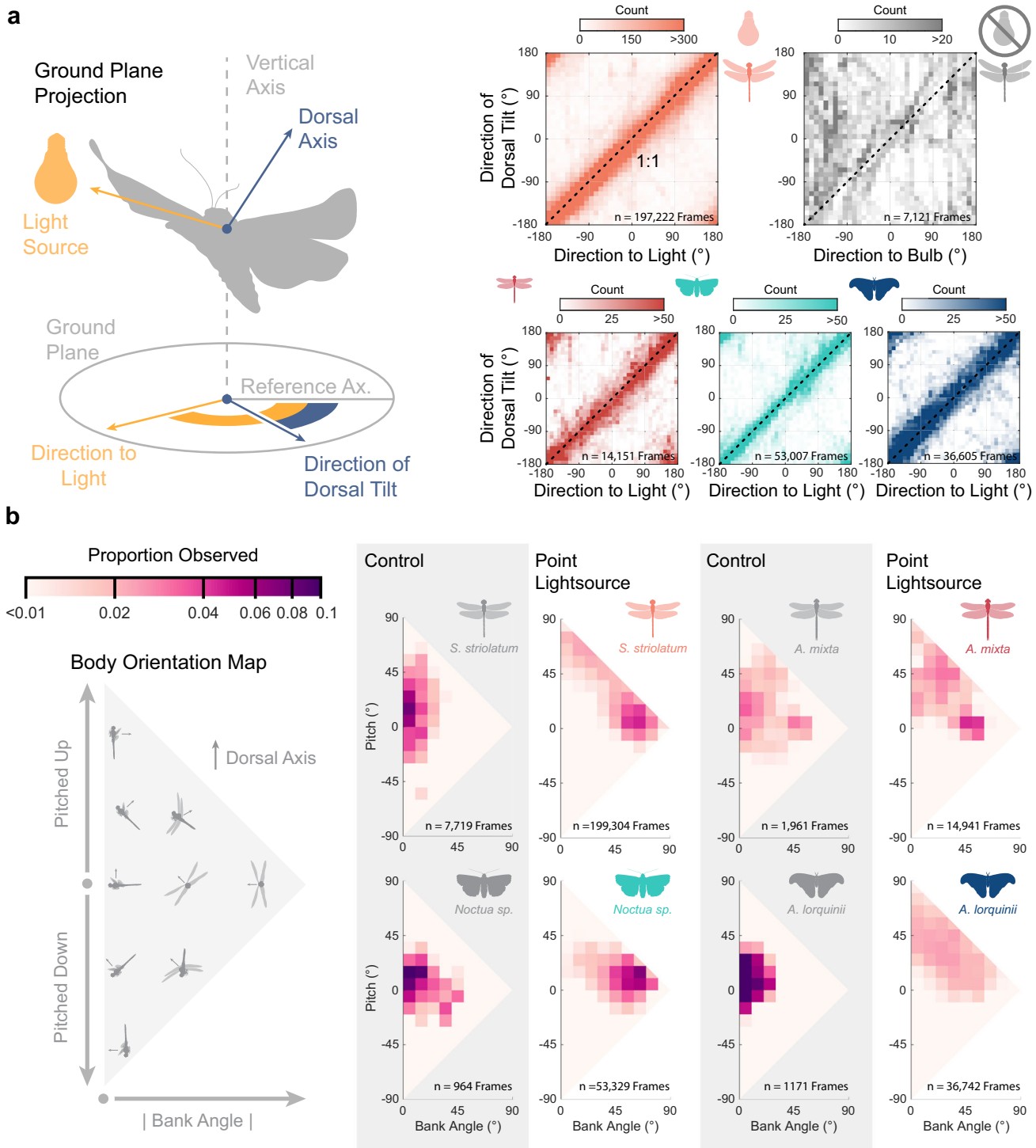

**Fig. 3 | Motion capture of the flying insects demonstrated that the animals maintain a consistent tilt of their dorsum towards the direction of the light.** **a** (*Left*) The insect's dorsal axis is projected onto the ground plane to compare with the light source direction. The reference axis is a global orientation reference. (*Right*) The direction of dorsal tilt is plotted against the direction to light. Dashed line shows a gradient of 1. Insects flying around a point source of light maintained extreme bank and pitch attitudes, as compared to animals flying under control conditions. **b** The relative body pitch and bank angle are plotted on a 2D distribution map. For each species, in-flight bank-pitch distribution under control conditions and near a point light source are presented on the left and right respectively.

measures of the 3D reconstructed trajectories (Supplementary Data 5). The total path tortuosity (total path length divided by distance travelled) for trajectories near light was higher (median 3.21, $n = 56$) around a point source than under a diffuse canopy (median 1.21, $n = 56$) (Wilcoxon rank sum $Z = 6.32$, $p < 0.001$) (Fig. 5a). Insects flying near point light sources tended to travel orthogonally (Rayleigh z test,

$z = 12.92$, $p < 0.001$, $n = 905$) to the light, an effect absent under the diffuse canopy ($z = 0.35$, $p = 0.70$, $n = 577$) (Fig. 5b). Finally, we tested for a light-centric turning bias when the light source was to the left or the right of the insect's velocity (within 30° of orthogonality when projected onto the ground-plane). Near a point light source, recorded insects preferentially turned toward the light ($X^2 = 114.66$, $p < 0.001$,

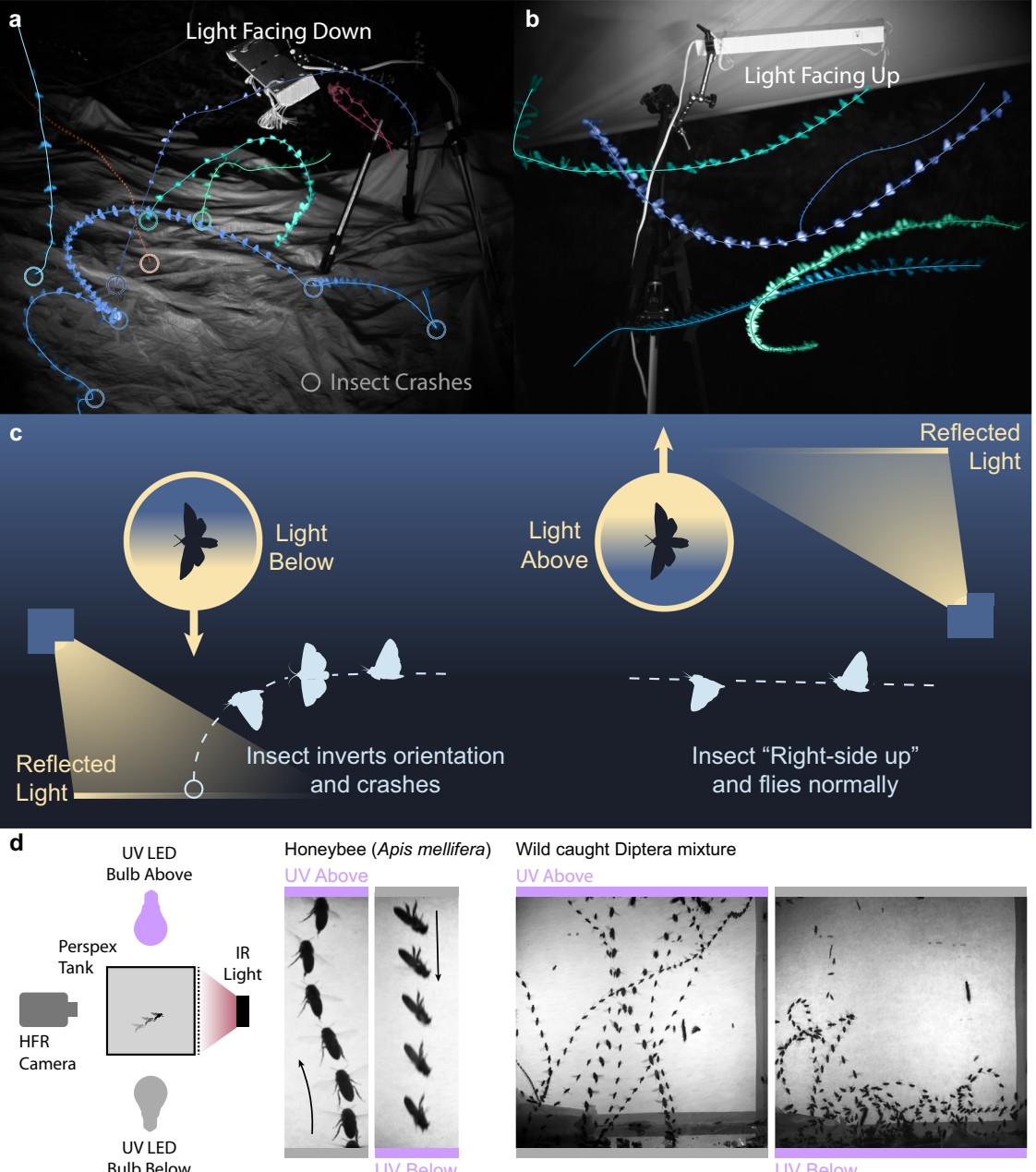

**Fig. 4 | The effect of reflected light was strongly dependent on whether it came from below or above the insect. a** Example trajectories of insects attempting to fly above a white sheet illuminated by a downward facing UV light tube. **b** Example trajectories of insects flying under a white sheet illuminated by an upward-facing UV light tube. **c** A diagrammatic representation of the hypothesised behavioural effect of 'light trapping' (*left*) vs. flight under a diffuse canopy (*right*). The strong effect of light directionality was also present in Honeybees and Diptera, both being unable to sustain flight when UV light came from below. **d** Example trajectories of Honeybees (every 30 ms), mixed wild Diptera (every 10 ms) flight with UV light provided above or below. Source data are provided as a Source Data file.

$n = 371$), as expected for a flight attitude in which they were tilted toward the light (Fig. 5c). This turning bias was absent under a diffuse canopy ($X^2 = 0.79$, $p = 0.37$, $n = 183$).

## Simulated dorsal tilting is sufficient to produce light entrapment

In simulation, we tested whether patterns observed under field and laboratory settings could have emerged from the proposed DLR mechanism alone. Due to anatomical constraints for flapping flight, flying animals often produce a net aerodynamic acceleration in a relatively constant orientation with respect to their body[37]. As a result, flying animals typically tilt their body to change direction, with the

exception during slow-flight manoeuvres (e.g. hovering)[33]. By reconstructing the aerodynamic acceleration (accounting for gravity) from our motion capture data, we found that the net acceleration vectors clustered within a narrow range forward and dorsal with respect to the insect's thorax (Supplementary Fig. 3).

Our agent-based simulations used a fixed acceleration vector relative to the insect's body axes (Fig. 6a). Maintaining flight requires the total lift to match or exceed gravity, and the forward component to counteract drag for the speed of travel. We used a linear proportional controller to construct this phenomenological model (see Methods for details). There were four free parameters – $k1$: the gain of dorsal tilting toward the light source, $k2$: the gain of corrective dorsal tilting toward

**Table 4 | Summary of small insect light direction flight assays (Total = 200 trials)**

| Order | Family | Genus | Individuals | Light from above trials | Light from below trials |
|---|---|---|---|---|---|
| Diptera | Muscidae | | 16 | 7 | 19 |
| Diptera | Anthomyiidae | | 12 | 14 | 17 |
| Diptera | Dolichopodidae | | 3 | 4 | 6 |
| Lepidoptera | Tortricidae | | 3 | 4 | 11 |
| Hymenoptera | Crabonidae | *Trypoxylon* | 2 | 3 | 3 |
| Ephemeroptera | Baetidae | *Cloeon* | 4 | 3 | 3 |
| Trichoptera | Limnephilidae | | 2 | 7 | 7 |
| Trichoptera | Leptoceridae | *Mystacides* | 5 | 7 | 10 |
| Hymenoptera | Ichneumonidae | *Ichneumon* | 2 | 3 | 4 |
| Hemiptera | Miridae | *Apolygus* | 1 | 1 | 1 |
| Lepidoptera | Crambidae | *Chrysoteuchia* | 3 | 5 | 6 |
| Hymenoptera | Apidae | *Apis* | 6 | 7 | 18 |
| Diptera | Drosophilidae | *Drosophila* | 21 | 8 | 22 |

true vertical, *k3*: the gain of stabilising the body axis towards the velocity vector, and *vt*: the terminal velocity of the insect acting as an index of drag magnitude.

Each of the three behavioural motifs (Fig. 1) were replicated in simulation by the model with the same parameter tunings given different initial position. With appropriate entry, the simulated insect developed an orbiting flightpath around the light with a stable flight speed over multiple seconds (Fig. 6b). Stalling was recreated by initiating the agent and flying away from the light source, with a steep light-induced climb and reduction in flight speed (Fig. 6c). Finally, when the agent's entry was initiated above the light source, it inverted its flight and entered a dive with rapidly increasing speed (Fig. 6d).

While the three motifs were generated with a fixed parameter tuning, the model assumption of light-induced dorsal tilting could readily induce light entrapment without specific tuning. We ran 300 five-second simulations with randomised parameters and starting positions (Fig. 6e). The majority of the simulated trajectories showed light entrapment via maintained or decreasing range from the light source. Removing the light-induced dorsal tilting (equivalent to turning the light off) showed the dispersal from the light source (Fig. 6f). We quantified dispersal under both conditions by measuring the average change in range to the light for the last 3 seconds of each simulation. Dorsal tilting models had a median range change of −0.06 m/s, not statistically different from 0 (Wilcoxon signed rank test Z = 0.76, *p* = 0.45), demonstrating that the agents did not escape from the light. Models without dorsal tilting had a median range change of 1.85 m/s, demonstrating dispersal away from the light (Wilcoxon signed rank test Z = 14.42, *p* < 0.001). As in experiments, average velocity direction of the simulated dorsal tilting models was orthogonal to the light, highlighting entrapment by a circuitous rather than direct path (Fig. 6g). Our model demonstrates that dorsal tilting is sufficient to generate flight paths that we observed in light entrapment.

We altered our simulation such that light response controller maintained the light at a fixed, but arbitrary, egocentric position (rather than purely dorsally). This model then represented a celestial compass that had been corrupted by the proximity of an artificial light source (Supplementary Fig. 4). Across 300 five-second random parameter simulations, the trajectories were a poor match to our observations of real animals. While some animals did spiral in toward the light source, trajectories lacked the consistent orthogonal-to-light trajectories observed in both real insects, and in the DLR simulations. Celestial compass simulations had a median range change of −1.75 m/s, demonstrating that agents escaped from the light (Wilcoxon signed rank test Z = 14.08, *p* < 0.001).

## Flight path manipulation via light switching

A corrupted compass cue could also result in insects travelling circularly around the light source (or more accurately in logarithmic spirals)[8,19,38]. To conclusively differentiate our flight control reflex hypothesis from the classic compass navigation theory, we toggled between two different point UV light sources while wild insects (see Table 5 for species composition) were orbiting beneath either light source (Supplementary Fig. 5, Supplementary Movie 6). We collected 70 monocular, upward-facing videos, of which 37 featured insects that orbited the new light source after switching. In the other 33 recordings, insects did not approach the second light source. Insects entrapped by the confusion of a celestial compass would endeavour to keep the perceived celestial object in the same relative position (left or right). However, we found that insects orbiting a light in one direction (e.g., clockwise) readily changed their side facing the light (swapped to anticlockwise) when we toggled light sources (insects swapped orbiting direction in 25 of 37 trials). Additionally, in 3 videos the insect switched orbiting direction on the same light source, without light switching. Dorsal tilting explains this rapid direction switching through body roll adjustment, which lacks the implicit L:R side constancy required for compass navigation.

## Exceptions to the light-entrapment behaviours

Some tested insect species seemed immune to light entrapment. Under laboratory conditions, none of the three Oleander Hawk-moths (*Daphnis nerii*) tested demonstrated light-orienting behaviour across 71 recorded trajectories. The hawkmoths flew directly above upward-facing UV and white LED bulbs without inverting their attitude or orbiting the lights (Supplementary Movie 7). The paths of *D. nerii* near the light lacked the orthogonal tendency seen in the other species (Supplementary Fig 6). The dorsal tilting index for *D. nerii* was 0.24 (*n* = 911), scarcely greater than that of *S. striolatum* in the dark. In general, *D. nerii* maintained a more level body attitude without the extreme bank and pitch angles seen in other species around a point light source (Supplementary Fig 6). Wild caught vinegar flies (*Drosophila* spp.) were another exception and showed no distinctive difference between flight above or below a UV or white LED light source (Supplementary Fig 6). These exceptions suggest that, in addition to the wavelength specificity, there are also species differences in this behaviour. Some species might not strongly rely on the light to correct their aerial attitude relative to the gravity.

## Discussion

We have used analysis of 3D flight trajectories to address the long-standing question of why insects aggregate around light at night

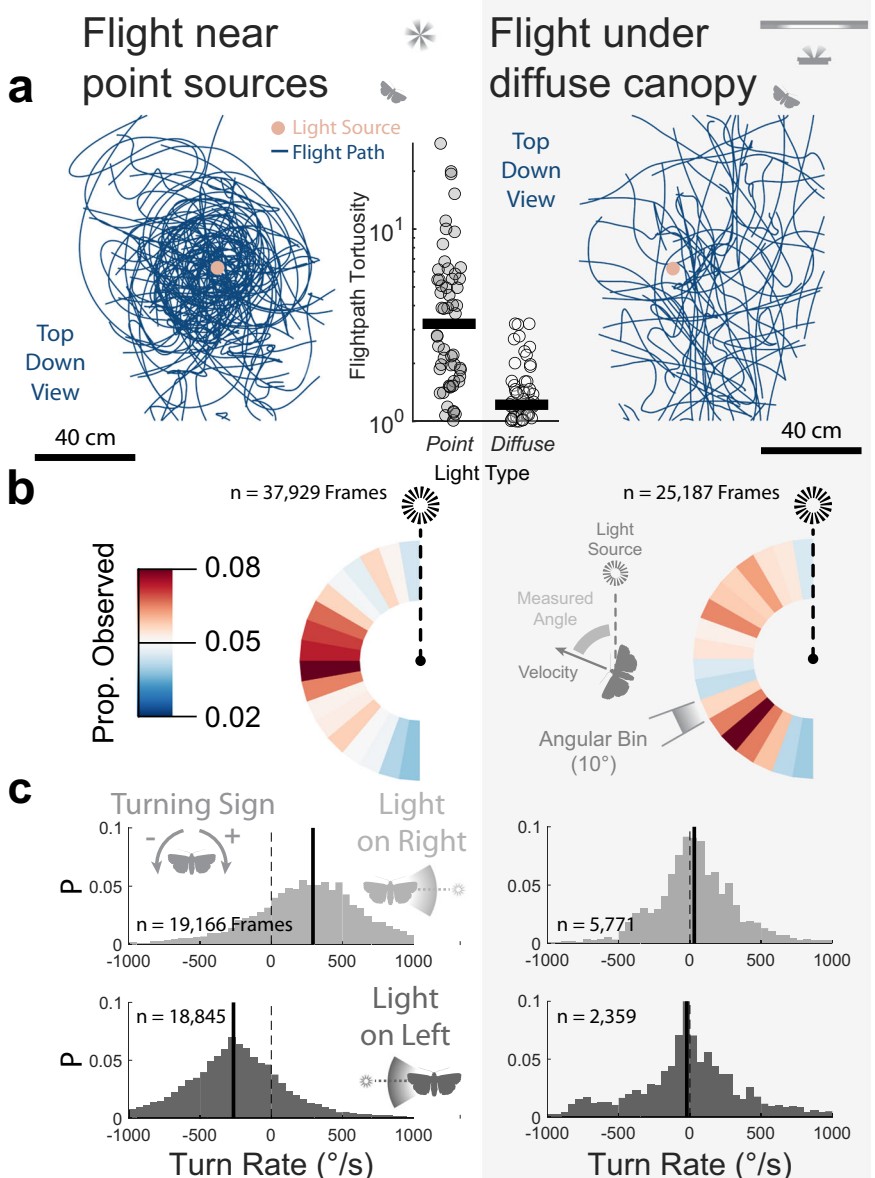

**Fig. 5 | Quantitative measures of insect flight around a point light source (left) and under a diffuse canopy (right).** Data presented from point sources came from randomly selected trials to match the sample size of the trajectories under diffuse canopy (*n* = 56 each). **a** As visible from their flight trajectories (viewed top-down), insects took tortuous circling paths around a point light source (*left*) and more direct flight under a diffuse canopy (*right*). The tortuosity of each trajectory is also plotted (*centre*). Insects travelling around a point light source predominantly travelled orthogonally to the direction of the light source, an effect not seen under a diffuse canopy. **b** Horizontal velocity orientation of insect flight relative to the position of the light source, coloured by the proportion observed. Insects also preferentially turned toward the direction of the light source when flying near a point light source, but not when under a diffuse canopy. **c** The horizontal turn rate distribution (positive for rightward, negative for leftward) for insects when the light (point-source page left, diffuse page right) was on their right (top), and on their left (bottom). Vertical bars indicate median values. Source data are provided as a Source Data file.

and seem unable to leave. We found that at short ranges most insects do not fly directly to a light source, but orthogonally to it, leading to orbiting, stalls, and even inverted flights. Qualitative observations from our field videos suggest that insects orient their dorsal axes towards light sources, and we confirmed this with insect motion-capture recordings in the laboratory. We propose a behavioural reflex model based on the well-documented dorsal light response of insects[26], arguing that a nearby artificial light source shifts an insect's sense of vertical orientation, disrupting its ability to maintain forward flight. Our experimental evidence and simulations attribute the mechanism of light entrapment to a disruption of the insect's perception of vertical rather than a navigational cue. We

discuss implications for this paradigm shift from navigation to control below.

### The moon and alternative explanations
We can now evaluate the previously suggested models with our experimental results. (1) Insects do not appear drawn to light as through an escape response[7]. In both field and lab conditions, insects rarely head directly towards, but consistently fly orthogonal to the light source. This refutes the fundamental premise of an escape response. (2) The confusion of a celestial compass by the light does not match our results either[8]. An insect should keep a light source at a fixed visual location for maintaining its heading. Switching light position

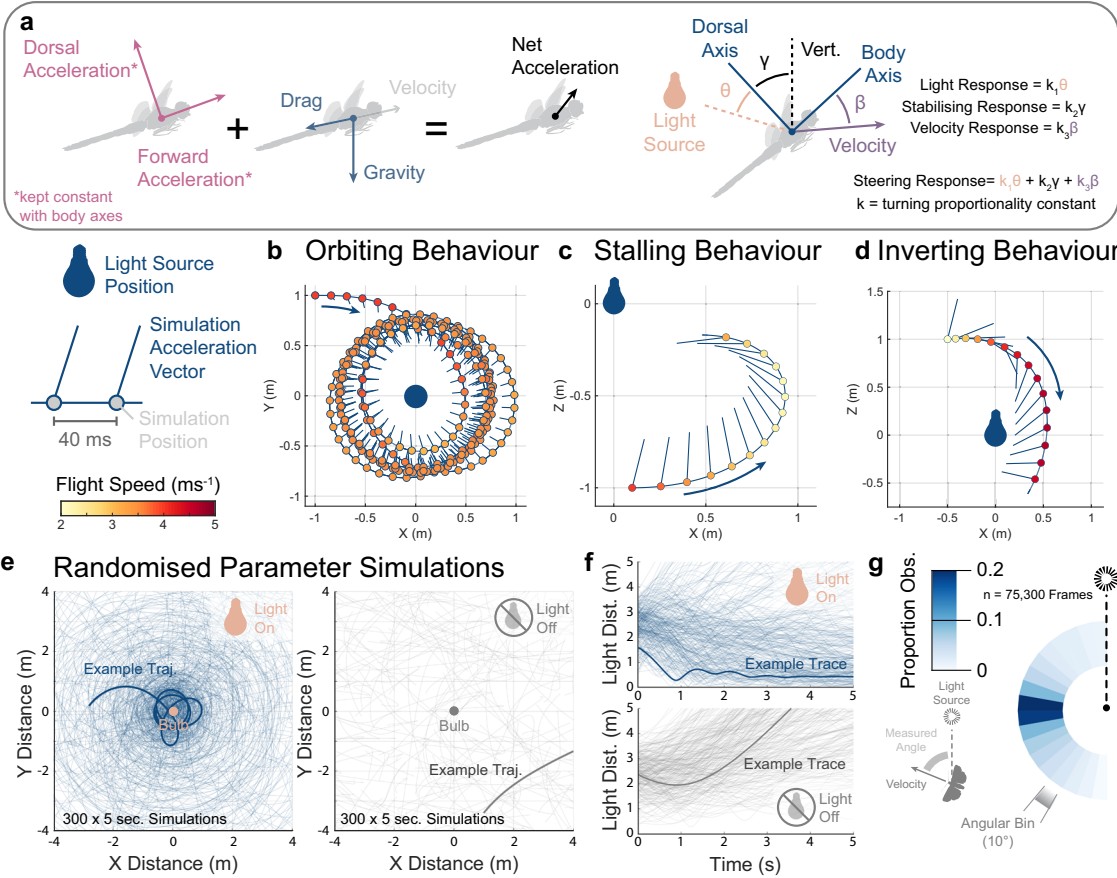

**Fig. 6 | Simulated insect flight when the direction of flight forces was limited by body orientation.** We used a proportional controller with three inputs to simulate the hypothesised dorsal tilting behaviour around a light source. **a** Diagrammatic representation of the net acceleration experienced by the simulated insect (*left*) and the simulated proportional steering controller rotating the body reference frame (*right*). We initiated the same simulated agent in three starting positions, generating the three behavioural motifs observed in the field. We have represented the simulated position and acceleration vector (combined dorsal and forward acceleration) of the agents by a point and vector line. Each point is colour-coded with the associated speed. **b** Simulated light entrapment behaviour during an orbiting motif viewed top-down. Flight speed is coded by scatter point colour. **c** Stalling motif model reconstruction, with track viewed side-on. **d** Inverting motif model reconstruction, with track viewed side-on. We randomised the model free

parameters within set envelopes to ensure the simulated light entrapment was not a product of exact parameter values. **e** Top-down plots of the trajectories taken by 300, 5-second, simulations with randomised free parameters. Simulations either tilted their dorsal axes toward the light (Light On, left) or were unaffected by the light (Light Off, right). A randomly chosen example trajectory is overlaid in bold for illustration. **f** Overlaid trajectories of the simulations' distance to light over time for dorsal tilting (*top*) and unaffected (*bottom*) trajectories. The same example trajectories as in **e** are overlaid in bold for both. Simulated agents predominantly travelled orthogonally to the direction of the light, as seen both in field and laboratory observations of real insects. **g** The orientation of the velocity vectors of the dorsal tilting simulations relative to the light source, coloured by the proportion observed. Source data are provided as a Source Data file.

(Supplementary Fig. 5) shows that insects readily hold the light source on either side of the body. We also do not observe logarithmic spirals toward the centre of the light source, a key prediction of celestial compass entrapment. A corrupted celestial compass cue also cannot explain why insects stall or invert themselves while flying over light sources[8]. (3) Heat radiation as an attractive component is refuted by the effect of LED lighting, which supplies negligible infrared radiation yet still entraps vast numbers of insects[9,39]. (4) Finally, the predictable light-centric flight trajectory motifs we elicited argue against insects being blinded by light[10,11]. Ultimately, we consider the dorsal-light-response the most parsimonious explanation of insect light entrapment. It is a basal sensory mechanism, thus explains the high prevalence of light attraction across a wide range of insects both diurnal and nocturnal.

**Some insects appear less affected by artificial light**
Among the insects we tested, only Vinegar Flies (*Drosophila* spp.) and Oleander Hawkmoths (*Daphnis nerii*) flew undisrupted over upward-facing ultraviolet light, but it is unclear why. Curiously, artificial illumination in the field readily entraps Oleander and other hawkmoths[40].

Additionally, there is evidence of mature hawkmoths foraging while ignoring bright artificial lights[41]. This implies state-mediated DLR suppression, or specific wavelength tuning across species.

The brightest visual region may be an overriding cue, but not the only cue for vertical orientation. During slow hovering flights, any mass hanging on the insect's body (such as legs) can indicate the gravity direction. However, gravity sensing via this method would be challenging during high-acceleration manoeuvres[28]. Combining optic flow cues and body rotation rate measurements may also enable an estimation of the gravity direction, as demonstrated in robotics[42].

Insects also fly when the zenith is not the brightest region, such as at dawn, on a forest edge, or when the moon is low in the sky[43]. Dorsal tilting towards the discrete natural light sources could lead to banked steering similar to that found around artificial light sources. One factor may be that insects adjust for the discontinuous brightness in the natural environment by local visual adaptation[44]. Currently, we do not understand why insects do not dorsally tilt toward natural celestial sources of light. In some species, the DLR has two components, mediated separately by the insects' compound eyes and by the ocelli[28]. Future work on the integration and luminance thresholds of these two

**Table 5 | Summary of light-switching trials in which an insect successfully switched from orbiting the first light source to the second (Total = 37 videos)**

| Order | Family | Total videos | Swap orbit direction | Same orbit direction |
|---|---|---|---|---|
| Lepidoptera | Pieridae | 20 | 13 | 7 |
| Lepidoptera | Nymphalidae | 3 | 2 | 1 |
| Lepidoptera | Noctuidae | 12 | 9 | 3 |
| Hymenoptera | Ichneumonidae | 2 | 1 | 1 |

components across multiple species would allow for a better understanding of how insects account for celestial light sources, and when artificial light destabilises them. Non-visual mechanisms like passive stability in insects capable of gliding should also help maintain a correct flight attitude without the requirement of sensory feedback[30], suggesting potential effects of body size on light entrapment[45].

### Long-distance attraction to light
We did not test the interaction between range and attraction, although other studies considered this in other contexts[46–48]. Other mechanisms might contribute to the arrival of insects at nocturnal light sources over longer ranges. For instance, insects do use celestial compasses for nocturnal navigation, and artificial light sources may interfere with these heading cues[8,49]. But even at long distances artificial light sources often remain brighter than the night sky and may cause dorsal tilting that would also steer an insect towards a light source. Only one experiment that we know of has tracked moth trajectories to lights over long distances, and found only 2 of 50 individuals released 85 m from a light source ended their flight their flight there[50]. This and our results suggest artificial lights may only trap passing insects rather than attract them directly from farther away.

Our findings suggest this light entrapment of insects at a local scale is due to a corruption of the insect's attitude control, rather than navigation. The DLR is a basal mechanism which enables vertical orientation. Bright lights can disrupt this mechanism and unintentionally alter insect flight. Taken together, reducing unnecessary, unshielded, upward-facing lights and ground reflections can mitigate the impact on flying insects at night, when skylight cannot compete with artificial sources. Future research focussed on spectral tuning of the visual components of the DLR would help isolate how best to alter artificial lights to avoid confusing insects flying at night.

## Methods
### Animal husbandry
The insects we used in lab experiments were either field caught (Yellow underwing moths, *Noctua pronuba* & *Noctua fimbriata* ($n = 8$, $n = 2$ respectively), Common Darter, *Sympetrum striolatum*, Migrant Hawker, *Aeshna mixta*) or reared from purchased pupae (Atlas Moth *Attacus lorquinii*, and Oleander Hawkmoth *Daphnis nerii*). All insects were kept on a 16–8 h light-dark cycle within a dedicated rearing tent at 24 °C and 65% humidity. Moths that fed as adults (not Saturniidae) were provided with halved organic bananas. Dragonflies were hand-fed adult *Drosophila* spp. during the few days in the lab.

### Artificial lighting
We provided experimental illumination by three alternative bulb types. The first was a blue-UV tube light common to insect light traps (Philips 15w TL-D Actinic, see Supplementary Fig. 7). The second a UV LED bulb (TBE Lighting L276, 9w, see Supplementary Fig. 7). Finally, we used a cold white LED light source (QNINE B22-G45 6000k, 6w). These lights were chosen to reflect a range of light spectrum that causes light entrapment in insects. However, we did not systematically pursue the effects of wavelength in this work. None of our light sources were

strongly polarised, negating any effects of attraction towards polarised light found in some insects[51]. Light sources were used in several different configurations within laboratory experiments. UV-Tube lights were always hung vertically in the centre of the arena. UV and White LED bulbs were placed centrally in the arena and either faced upward (bulb up), or downward (bulb down). To create control light-environments, we used either broad spectrum HID lamps (2 x Philips CDM-TMW Elite 315 W diffused through the tent ceiling) for diurnal species, or a single shrouded UV tube shining onto the ceiling to create a low-intensity UV canopy for nocturnal species.

### Field stereo videography recordings
We made field recordings (Supplementary Fig. 1, Supplementary Data 1) at Estación Biológica Monteverde (EBM) and CIEE, Monteverde Field station, Costa Rica, under permit numbers M-P-SINAC-PNI-ACAT-024-2020 and R-SINAC-ACG-PI-016-2022 issued by SINAC (National System of Conservation Areas). Data was collected in two separate field trips (Jan-Feb 2022 and May 2023). We used a pair of monochrome shutter-synchronised Fastec TS3 high-speed cameras mounted on a single tripod cross-arm. Most videos were shot at 500 fps, giving a good temporal resolution for flight behaviour. These cameras permitted us to film with infrared illumination, which we assumed invisible to the insect eyes. Consistent with this assumption, we did not observe any insects crashing into, nor interacting with our IR lights (Larsen wide angle IR Illuminators; 850 nm). We configured the IR lighting to create high contrast for flying insects against the dark night background. While the exact camera orientations and distance from the light-source varied from night to night, we invariably centred the light source in the field of view of both cameras. In practice, this gave us a maximal recording volume of 1.5 m x 2 m x 1.5 m (Width x Depth x Height, with height aligning with gravity). To provide stereo calibration, we waved a known-sized checkerboard through the overlapping views of both cameras. We could then use the inbuilt MATLAB camera calibration app (Computer Vision Toolbox 10.3) to both detect the checkerboards in the views of both cameras and estimate both the intrinsic (optical centre, focal length, and radial distortion) and extrinsic camera parameters (relative camera orientation and translation). Within our field recordings, we were unable to identify many insects below order-level with certainty. We filmed the lights in several different configurations. **'No Light'**: The cameras were pointed at a region of space in the forest without any illumination from UV or visible bulbs or tubes. IR illumination was still provided and minimal leakage in the red was present (Supplementary Fig. 7). **'UV Tube'**: The UV tube light was suspended vertically from rope between two trees ~1.5 m from the ground. Macro UV Tube is the same treatment, but to observe insects the cameras were moved closer for a few trials, for the purpose of data analysis both treatments have been combined. **'UV Bulb Up'**: The UV LED bulb mentioned above was affixed to a tripod ~1 m from the ground pointing upwards. **'UV Bulb Down'**: The same UV LED bulb was suspended ~1.5 m from the ground. **'White Sheet Above'**: A white cotton sheet was suspended ~2.5 m above the ground and the UV tube was pointed upwards close to the light using an extra shielding at its base to prevent non-diffuse downwelling light. **'Sheet Down'**: The white cotton sheet was spread on the forest floor and the UV tube was kept close to the ground facing downwards ~0.3 m from the ground. To get a better overview of the different insects coming to light and to increase the sampling of species, we repeated light attraction to experiments with an additional round of fieldwork in 2023 using a known set of 30 insect species, spanning 6 orders with a single light condition (UV Bulb Down). Insects were photographed to confirm their identity and then were released and filmed with the stereo setup described above.

We qualitatively surveyed the videos noting the different motifs observed (presence-absence) and the taxon present in the video (identified to different taxonomic levels). **'Orbit'**: Animals

travelling in an arcing pattern around the light (even if the loops were not complete) were classified as showing orbiting behaviour. From reconstructed trajectories, looping behaviour was easy to identify from the circular or oval paths when viewed from above. **'Stall'**: Animals flying upward and slowing down while pitching upward but not completely inverting was classified as showing stalling behaviour. **'Invert'**: Animals tilting their dorsum full downward for any portion of the flight were counted as invert.

In several videos, multiple motifs were present at different parts of the flight and in different species, making it hard to quantify, and thus we included any of the motifs seen across all animals in the video. The final classification is somewhat subjective, but we provide the original videos in the final repository for future cross-verification. We also repeat this for the digitised tracks, where the motifs are identified for only the tracked insects in each track.

## Insect Marking for Motion Capture in the Laboratory

Our motion capture system relies on retroreflective markers affixed to the recording subject (Supplementary Fig. 2, Supplementary Data 2). To mass produce retroreflective marker frames (3 markers per frame), we used a stereolithographic 3D printer (Formlabs Form 3). We then added small (1 mm$^2$) sections of adhesive retroreflective tape (Qualisys) to the spherical markers. The resulting photo-polymeric resin marker frames were slightly heavier than the carbon frames we used previously[30] (10–20 mg per marker set), but with much reduced fabrication time. For the flight behaviour of this study, this weight still had minimal impact on the flight at ~5% of bodyweight for our lightest insects (*S. striolatum* at ~300 mg). After the subjects were immobilised on ice, we attached a marker frame to the dorsal surface of the thorax using a minimal amount of UV curable glue (Loctite 4305). A custom UV LED pen with a small light window (3 mm) was used to cure the glue locally to minimise any risk of damaging the insect's vision. All insects recovered in the behavioural tent for 20 min before we began recordings. We found no visible behavioural differences between the marked and unmarked insects, suggesting that the marker frame did not impact the general flight control.

## Motion capture behavioural recordings

We used eight Qualisys Marqus M5 motion capture cameras (4 pairs) recording in infra-red (850 nm) arrayed around a steel ring (diameter 1.66 m) (Supplementary Fig. 2). This was held on a vertically movable metal frame (2.4 × 2.4 m) suspended from the ceiling and both raised and lowered by a central winch. We used blackout curtains to prevent stray light (e.g., computer screens) in the laboratory from affecting our results. From this same frame hung a white cylindrical tent (diameter 2 m, height 2.4 m) with the cameras poking through portholes near the ceiling. The tent was composed of white Joelastic fabric (J & C Joel) and the reflectance spectrum from the UV actinic tube light can be seen in Supplementary Fig. 7. Lights were hung in the middle of the tent 1.5 m above the floor, allowing insects to fly freely around them. Flights were either spontaneous, or manually elicited by brushing the insect's abdomen. Multiple flights occurred within the same recording, and each recording ran for a maximum of 30 min. During these recordings, the insect was free to leave the cameras' view, and then return. We optimised the motion capture recordings to maximise the covered volume and recording length. The covered recording volume took the shape of a cylinder 1.6 m in diameter, 1.5 m tall, with the light-source at its centre. However, reconstruction depended on consistent marker visibility, which varied with distance from the light (and thus from the centre of the recording volume). As a result, contiguous stretches of reconstructed flight were generally within 0.5 m of the light source. This configuration provided a tracking residual ~0.24 mm at 240 fps (Supplementary Fig. 2).

## Field data processing

Our field data were more variable than the laboratory data, and some videos did not yield usable trajectories. One major disturbance was the wind. We chose filming sites that were sheltered from the wind as suggested by an anemometer, and tried to record when the wind speed was under 1 m/s. However, wind speed relative to the ground varies widely over both time and space at a scale relevant for the recordings. Thus, we could not estimate the true airspeed of the insects with high certainty. Smaller insects are likely to be more affected by airflow due to their lower mass and slower flight speeds. Even low wind velocities may have impacted the flight patterns we observed in the field. In any case, we processed all flight trajectories that were resolvable and not visibly impacted by the wind.

Another source of field data variation was image digitisation error. Our data processing pipeline was developed for insects with high IR reflectivity, thus insects which reflected less IR were difficult to track. For example, the dorsal light response was robustly discernible in clear wing butterflies, however the transparent wings made accurate digitisation impossible. Similarly, most of the insects visibly present at our light were small (<1 cm body length) yet in our data the mean insect size was 29 mm ± 9 mm (estimated via angular size and distance from the cameras). This bias towards larger insects was due to IR reflection visibility in the recorded footage.

We created three custom MATLAB apps to assist with the digitisation and triangulation of field data, their function was as follows: (1) Identify and label the beginnings of trajectories (start indexes) in both camera views, obtaining the start frame and position of multiple trackable paths within a single set of paired videos. (2) Import both videos and the trajectory start indexes. Then build a smoothed spline by scrubbing through the video and adding position nodes on the tracked insect's location (every 50–100 frames). Tracking could then be applied by subtracting an averaged background frame (obtained from 20 linearly spaced frames throughout each video) from each frame along the insect's track. On each frame, the app created a search box around the interpolated spline and searched the binarized subtracted image, locating the focal insect by its proximity to the tracking spline. The light source, if there was one, was also digitised within this app. (3) Finally, the raw position measurement of the insect was triangulated from the tracked insect centroids and the recorded calibration for that selection of videos. The nature of the tracking meant that high-frequency oscillations were created by the centroid focussing on the wings of the tracked insect, these were counteracted by fitting a cubic smoothing spline through the obtained track. We used a smoothing constant that maintained the course of the insect within the bounds of the oscillations created by the wingbeats to avoid over-smoothing. This gave a smooth estimation of the position and velocity of the insect during its flight.

## Motion capture data processing

Motion-captured markers were labelled in the proprietary Qualisys Track Manager software and then exported directly into MATLAB structures. Markers were identified by their asymmetric placement, but secondarily filtered based on their known distance to other markers. Two quality filters were applied to the tracked data to ensure accuracy. (1) If the distance between the markers on either arm of the frame exceeded 0.4 mm of the median length (10% of the length of the shortest marker arms) or (2) the angle between the arms differed by > 5° from the median (~90°), the frame was removed from the trajectory. These instances reflected either poor tracking or accidental mislabelling.

Recordings of up to 30 min consisted of many smaller sections with variable marker visibility. Individual trajectories were excised if the marker frame was not visible for longer than 0.5 seconds. This kept closely time-linked trajectories together despite small gaps but separated different bouts of flight around the light. When analysing the data,

we used separate flags to distinguish 6 DoF data (in which all three markers of the rigid-frame were tracked) from 3 DoF data in which only one marker needed to be visible. While we required 6 DoF data to distinguish orientation of the insect, we could still use 3 DoF data to demonstrate flight speed and the position of the insect around the light. Given that insects would frequently settle on the light or walls and occasionally walk around at low speeds, we filtered out any data below 0.3 ms$^{-1}$ to avoid including data in which the insects were not in flight. To quantify the insect's body orientation, we used a composite of non-additive bank and pitch angles relative to the horizontal plane. For pitch angle, we measured the angle between the animal's long body axis (from posterior to anterior) and the global vertical. To quantify bank angle, we calculated the magnitude of the angle between the insect's lateral vector (aligning left to right laterally across the animal's body) and the horizontal plane. It should be noted that this methodology reflects around the horizontal plane (maximum bank is limited to 90°), meaning that an insect completely inverted upside down would score the same as one the correct side up. For our motion-capture recordings, this was not of practical relevance, as near-inverted animals would obscure their markers from the camera system above.

### Laboratory video recordings

To test the effects of artificial light on smaller insects than those used in our motion capture recordings, we caught Honeybees (*Apis mellifera*) and an assortment of Diptera & other Hymenoptera from the grounds of Imperial College London. We also collected small insects arriving at a light trap in Cambridge, UK. These crepuscular/nocturnal taxa included Trichoptera & Ephemeroptera (see Table 4 for full taxonomic breakdown). Captured diurnal insects were recorded in experiments within 1 h of capture (14:00 to 18:00). Nocturnal light trapped insects were recorded the following day within 24 hrs. We collected *Drosophila* spp. from a local compost heap (Cambridge, UK), using them within 48 h of capture. *Drosophila* spp. were given small sections of banana on which to feed before recordings were made.

We contained these insects within a Perspex-sided cube 20 cm on a side (Fig. 4d). A small portion of damp cartridge paper in one corner of the box provided sufficient humidity that most tested insects survived the experiments and were able to be released afterward. On one side of the cube, we placed two infrared LED panels facing through the centre of the box (850 nm Splenssy 96 LED array). The closest wall of the cube was covered in thin paper, diffusing the transmission of the infra-red light to create a near-even backdrop against which insects could be silhouetted. We placed UV (TBE Lighting L276, 9w) or white LED bulbs (QNINE B22-G45 6000k, 6w) above and below the cube, having independent control of each via toggle switches.

We positioned a high-speed camera (Phantom v211, Vision Research, with 50 mm Nikon F-mount lens) to look through one wall of the cube at the diffuse infra-red illumination. We recorded the flight behaviour at 1000 fps. Switching between the lights caused insects to congregate either at the top of the container (when the light came from above) or at the bottom (when light came from below). Periodically switching between the lights was generally sufficient to elicit flight responses. We also found lightly tapping the box a reliable method for generating flight recordings, especially in *Drosophila spp.* who did not congregate around the light source.

### Light switching

For our light switching experiments, we hung two UV LED lights (TBE Lighting L276, 9w) from a metal frame (3 m tall) outdoors in Cambridge, UK (Supplementary Fig. 5, Supplementary Movie 6). The lights were thus suspended 2 metres from the ground. We arranged a single high-speed camera (Chronos 2.1, Kron Technologies, with IR filter removed) facing directly upward beneath the lights. Either side of the camera we arranged two IR illuminators (850 nm Splenssy 96 LED

array) facing upward, which picked out flying insects against the dark sky above. We recorded behaviour at either 500 or 250 fps.

We switched on one of the lights and waited for wild insects to begin orbiting behaviour. We also introduced wild-caught diurnal species by releasing them individually. When an insect was orbiting beneath one of the lights, we swapped to the other light using a manually toggled switch. After a short interval (<5 s) we manually triggered the camera and saved the video.

### Data analysis

All behavioural kinematics and analyses were produced in MATLAB 2021a (MathWorks) using custom scripts. Example scripts are provided along with the flight data themselves.

### Statistical methods

We performed all hypothesis tests in MATLAB 2021a (MathWorks). We used Bonferroni correction to adjust our threshold of statistical significance for all hypothesis tests (0.05/18 = 0.003).

When testing sequential samples from trajectories gathered at high frequency, we considered that individual frames could represent pseudo-replicates. To counteract this, we subsampled trajectories to 10 Hz (e.g. for 500 fps data, we sampled every 50$^{th}$ frame). We chose this because it allows for sampling at a frequency that allows the insects to change bank and pitch angle (which often vary at rates >500 °/s) substantially between samples, while retaining sufficient samples for statistical power.

For testing the nonuniformity of our circular distributions (e.g., for horizontal velocity relative to the light) we used the Rayleigh z-test. This test assumes that non-uniformity is unimodal. However, when concerning insects travelling or tilting around a light source, our expected distributions are bimodal (insects can travel both clockwise and anticlockwise around the light whilst orbiting). To correct for this effect, we used an angle doubling procedure. All velocity bearings from the light (0°–360°) were doubled. We then subtracted 360° from any doubled angles >360°. This resulted in the bimodal clockwise-anticlockwise orbiting forming a single unimodal concentration.

When testing the turning direction of insects near point sources versus those near diffuse sources, we used a chi-square test. For each condition, created a contingency table with two variables: light direction (left or right), and turning direction (left or right). Trajectory data from each condition was subsampled to 10 Hz, as in other trajectory analyses.

### Light and environmental measurements

We measured the spectra of the two UV lights (LED bulb) and the UV Tube used in the field and lab experiments. We also measured the spectrum of the reflected light inside the laboratory tent. We used a calibrated FLMT03251 Flame Ocean Optics Spectrophotometer to take relative irradiance measurements. An Integration Time of 40 ms, with 10 Scans to average and electric dark correction enabled, and no nonlinearity correction enabled, with a Boxcar width: 0 were used. The light sources were placed a meter from the light and tilted to ensure the sensor was not saturated. For the dark sky measurements, we used the Environmental Light Field setup (https://github.com/sciencedjinn/elf) as described in[52]. We used a Nikon D850 with a Sigma 8 mm/F3.5 lens to take a dark calibration as recommended with a 20 s exposure time. A Govee H5072 humidity and temperature hygrometer was used to take measurements in the 2021 field experiments. Wind speed was recorded using a handheld anemometer. The Ambient Weather WS-2902 was used to measure humidity, temperature, and wind speed for the experiments in 2023.

### Simulating dorsal tilting

To mimic the dorsal turning responses, we introduced a proportional controller that pulled the dorsal axis of our simulated insect

towards the direction of the light source (Fig. 6a). The proportional controller caused the insect's body to rotate with an angular speed proportional to the error between the dorsal axis and the line-of-sight (LOS) vector to the light source. We termed the gain on this controller $k_1$, in units of s⁻¹. We did not attempt to capture realistic flight dynamics but merely to provide an approximation of the observed dorsal tilting phenomenon. We implemented a second controller to pull the dorsal axis back towards the true vertical with a gain $k_2$. This stabilising controller represented active and passive mechanisms in the insect's flight system that may orient the insect right-side-up. We included this based on our observation that *S. striolatum* flying in total darkness still retains some degree of correct body attitude. Finally, we introduced a third controller that pulled the longitudinal body axis of the insect towards its velocity vector. This accounts for the active and passive effects of an insect's body tending to remain head-on to its direction of travel. Thus, the planar formulation of the steering embodied by the simulated insect is given by:

$$\dot{\theta} = k_1\theta$$
$$\dot{\gamma} = k_2\gamma \qquad (1)$$
$$\dot{\beta} = k_3\beta$$

Where $\theta$ is the angle between dorsal axis and the LOS to the light, $\gamma$ is the angle between the dorsal axis and vertical, and $\beta$ is the angle between the body axis and the velocity. $k_1$, $k_2$, and $k_3$ are the respective proportionality constants for the steering responses. $\dot{\theta}$, $\dot{\gamma}$, and $\dot{\beta}$ are the angular velocity steering corrections to the simulated body axes due to the light source direction, passive stability, and velocity direction respectively. The effects of these corrections were summed within each model time step.

**Drag**

The aerodynamic drag for flapping insects depends on multiple influences, including speed, wing posture, and body orientation amongst other factors. Here, we adopted a simplistic quadratic air drag model with a form factor $c$. This constant could be determined by setting the terminal velocity achieved by an insect in freefall. Varying the terminal velocity of the simulated agents allowed for the characterisation of insects of differing sizes.

Where c is a constant reflecting the deceleration due to drag for a given airspeed, $g$ is gravitational acceleration (scalar), and $v_t$ is the insect's terminal velocity (scalar). Within our simulations of the three behavioural motifs, we used a constant of $0.80$ s⁻¹ with a terminal velocity of 3.5 m/s. This value was chosen as it kept simulated flight-speeds similar to those measured in our motion-capture recordings. During random gain simulations, we set the constant between $0.09$ s⁻¹ and $39.24$ s⁻¹ (terminal velocity between 10.5 and 0.5 m/s² respectively).

$$c = \frac{g}{v_t^2} \qquad (2)$$

**Kinematics**

Simulations were run on a discrete time interval basis. The evolution of the flightpath being governed by the following set of equations. Vectors are written in bold. Firstly:

$$\boldsymbol{a_b} = \left|\boldsymbol{a_{forw.}}\right|\widehat{\boldsymbol{b_x}} + \left|\boldsymbol{a_{dors.}}\right|\widehat{\boldsymbol{b_z}} \qquad (3)$$

Where $\boldsymbol{a_b}$ is the acceleration generated by the model insect. $\boldsymbol{a_{forw.}}$ and $\boldsymbol{a_{dors.}}$ Are the forward and dorsal components of the generated acceleration, and $\boldsymbol{b_x}$ and $\boldsymbol{b_z}$ are the x and z axes of the insect's body. The hat notation over the vectors denotes unit vectors. The insect's

body rotates based on the controller described in the 'Dorsal Tilting' Section. This acceleration is then combined with gravity and drag to create the net acceleration.

$$\boldsymbol{a_{net}}(t) = \boldsymbol{g} - \hat{\boldsymbol{v}}(t-1)(c\left|\boldsymbol{v}(t-1)^2\right|) + \boldsymbol{a_b}(t) \qquad (4)$$

Where $\boldsymbol{a_{net}}$ is the net acceleration on the simulated insect's body. $\boldsymbol{g}$ is gravitational acceleration, $c$ is the drag constant, and $\boldsymbol{v}$ is the velocity vector of the simulated insect. Here, $(t-1)$ refers to the value from the previous time step. This net acceleration is then added to the body kinematics as follows:

$$\boldsymbol{v}(t) = \boldsymbol{v}(t-1) + \boldsymbol{a_{net}}(t)\Delta t \qquad (5)$$

$$\boldsymbol{p}(t) = \boldsymbol{p}(t-1) + \boldsymbol{v}(t)\Delta t \qquad (6)$$

Where $\boldsymbol{p}$ is the position of the simulated insect and $\Delta t$ is the elapsed time between iterations.

**Flight simulations**

There are 4 model parameters to set in the simulation. To recreate example motifs observed in nature, we chose the following parameters: $k_1 = 15$ s⁻¹, $k_2 = 1$ s⁻¹, $k_3 = 15$ s⁻¹, $v_t = 3.5$ ms⁻¹. The forward component of acceleration was 5 ms⁻² and the dorsal component was 15 ms⁻². For these examples, the $\Delta t$ per iteration was kept at 10 ms. These parameters were chosen to reflect an insect flying at a relatively low Reynolds number (low terminal velocity), rapid aerial mobility (k values giving rapid reactions like those measured in insect pursuit flight controllers), and with lift and thrust profiles like those observed in our measured data (Supplementary Fig. 3).

To avoid conclusions drawn from a well-tuned combination of parameters, we initiated 300 simulations with randomly assigned parameters within reasonable ranges. The ranges were as follows: $0$ s⁻¹ $< k_1 < 20$ s⁻¹, $0$ s⁻¹ $< k_2 < 20$ s⁻¹, $k_3 = 15$ s⁻¹, $0.5$ ms⁻¹ $< v_t < 10.5$ ms⁻¹, $0$ ms⁻² $< \boldsymbol{a_{forw.}} < 10$ ms⁻², $9.81$ ms⁻² $< \boldsymbol{a_{dors.}} < 24.81$ ms⁻². We determined starting positions at random within the cube defined by +/− 5 m of the light source along each of 3 spatial axes. Initial headings were parallel to the ground plane but started at a random horizontal bearing. Although interactions over the full parameter set are outside the scope of this work, we found the agents in most simulations were entrapped, drawn closer to the light with many entering a stable orbit.

Additionally, we adapted our flight simulations to match the assumption of a corrupted celestial compass. This model used the same structure as our dorsal tilting model, but with the light response component of the controller attempting to maintain the light at a fixed egocentric direction (rather than over the dorsum). This arbitrary direction was set by the initial direction of the line-of-sight to the light source, from an egocentric perspective. All other components of the simulation were kept the same as previously discussed.

**Reporting summary**

Further information on research design is available in the Nature Portfolio Reporting Summary linked to this article.

## Data availability

The raw stereo-videos and processed trajectories associated with this manuscript are available via Figshare (https://doi.org/10.6084/m9.figshare.24771978). Example high-speed videos of our experiments are provided in Supplementary Movies 1–7. These data include both the video-tracked 3D trajectories and 6-DoF laboratory motion capture trajectories of insects around light. All processed data used to make figure panels is available in a source data file. The source data file also contains the all the required data to replicate our statistical testing of hypotheses. Source data are provided with this paper.

## Code availability

The code required to handle both field and laboratory data is available via Figshare (https://doi.org/10.6084/m9.figshare.24771978). Example data handling scripts are provided for each recording type and allow for the replication of our results. This repository also includes the flight model used to simulate the effects of dorsal tilting.

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

## Acknowledgements

We thank the CIEE Monteverde staff for logistical support and Marvin Hidalgo for logistical support at The Monteverde Biological Station. We thank Enrique Castro for assistance with permits and Tim Brandt for help troubleshooting the hi-speed camera setup. We thank the staff members of the Institute of Environment and the Kimberly Green Latin American and Caribbean Center for help with logistics and grant management. Dr. Alexander Yarger for discussion and feedback on the results. Dr. Akito Kawahara, Prof. Andrew Biewener, Dr. Aso Yoshi and Prof. Holger Krapp, for feedback and comments on the manuscript. We thank Dr. Nathan Lord for help with light measurements. We thank Labonte Lab for use of their Form 3 SLA printer. We would like to thank Dr. Eleanor Miller for assistance obtaining and housing research subjects. This paper is contribution #1667 from the Institute of Environment at Florida International University. Financial support was provided by the European Research Council (ERC-StG no.804315 'Vision-In-Flight' to HTL), National Science Foundation (NSF IOS-1750833 to JCT), and U.S. Air Force Office of Scientific Research (AFOSR MURI award FA9550-22-1-0315 to JCT). Additionally, YS received support from a DYF award from the FIU Graduate School. Fieldwork was supported by the following: a Tropical Conservation Grant from the Susan Levine Foundation, a National Geographic Explorer Grant (EC-82941R-21 to YS & STF), a Lewis Clark Exploration Grant from the American Philosophical Society, and a Tinker Field Research Collaborative Grant.

## Author contributions

S.F. conceptualization, methodology, data gathering, software, visualisation, writing–original draft, data curation, formal analysis, project administration, fieldwork funding acquisition. Y.S. conceptualisation, methodology, data gathering, writing-original draft, data curation, project administration, fieldwork funding acquisition. J.T. project supervision, visualisation, data gathering, analysis validation, writing-review and editing, project administration, funding-acquisition. H-T.L. project supervision, simulation advising, analysis validation, writing-review and editing, project administration, funding-acquisition. P.A. fieldwork support, writing-review and editing, project administration.

## Competing interests

The authors declare no competing interests.
