## [Peer Review File · Nature Communications]

Why flying insects gather at artificial lightReviewers' Comments:

Reviewer #1:

Remarks to the Author:

The authors tackle a question that humans have wondered about for thousands of years—why insects are attracted to light. Despite numerous hypotheses, there is surprisingly little data available in the literature. The authors tackle this with a combination of field and laboratory experiments and flight simulations. They present what appears to be an entirely novel hypothesis regarding why insects are attracted to light—corruption of a dorsal-light response for determining vertical orientation. This is similar, but crucially, quite distinct from the classic hypothesis that insects hold bright lights at a fixed orientation for nocturnal navigation.

Overall, the experiments are quite convincing of the proposed hypothesis and negate several other proposed hypotheses. The number of experiments conducted and the detail regarding the analyses conducted are quite impressive. In most cases the results show very strong effects. I do have two small points that I would like to see addressed:

1) In some cases—specifically the first field experiments—there is insufficient data provided. For example, how often was each behavior observed? Also what were the insects that were observed? I realize it is not always possible to identify the species of insects, but the authors should do the best they can to identify at least the order.

2) In many cases the authors present very convincing data, but fail to analyze it statistically (for example, the plots in figure 5c). I have noted several places below where statistical analyses should be conducted in addition to the descriptive statistics presented in the current manuscript.

Overall, I consider this a ground-breaking piece of research—one of the most important papers I've seen published in this field in recent years. It has numerous implications for insect biology, conservation and behavior and provides clear data to answer a question humans have pondered for millennia.

Line-specific comments

Line 87 – Please provide some summary statistics regarding the duration of these recordings.

Lines 86-99 It would be helpful to quantify the frequency of these behaviors. How often was each identified? Were there any trends in what insects exhibited which behaviors more frequently relative to insect order or body size? Also, what were the insects that you recorded? Can you at least provide a summary based on your ability to identify the insects to order or family level when possible?

Lines 114-115 Again, what were the durations of these recordings?

Lines 120-123. The circular histogram plots look convincing, but you should also conduct statistics on the distributions, as currently in the figure all of the plots, including your control, have medians close to 90 degrees.

Lines 148-150. These values should be compared statistically.

Lines 167-168. Are these insects active during the day, night or both?

Lines 182-183. Circular statistical tests should be conducted to demonstrate whether each distribution overlaps with zero.

Line 235. Here, you need to report the number of trials that were conducted and the number of times each moth switched orbiting direction after the light switch.

Line 264-265. What are you basing this statement on? You didn't show any data of dorsal orientation of moths in the field, so is this just based on your qualitative review of the videos from the field?

Line 326-328. What about the reflectance of the ground upon which lights are shining? Even if lights are pointed down, if the ground is well lit and reflective, would that cause insects to be attracted and make inversions such as the ones you demonstrated?

Line 362 – Should also mention how the camera intrinsics were determined.

Line 456 – Looks like a typo, check wording.

Line 500 – citation here would be helpful.

Lines 518-519 – How were these constants selected?

Line 535, dt should be replaced with Δt , since the change is not infinitesimal.

Figure 6. Perhaps I missed this somewhere, but if you haven't already, please describe what the

"sticks" of the "lollipops" represent in panels b-d.

Sincerely,

Aaron J. Corcoran
University of Colorado, Colorado Springs

Reviewer #2:

Remarks to the Author:

Review of NCOMMS-23-09906-T

Dear authors,

I read your manuscript on your proposed explanation for why insects gather at artificial light and think that in general your work does a good job proposing and supporting a novel and interesting explanation for a widely observed phenomena of general interest. I do have some broader questions as well as some smaller notes that should be addressed.

One question that immediately came to mind was whether the size of your 3D tracking volume for the in-lab experiments permits any estimation of the "capture radius" of the lights you were using, or whether the data show any variation in general with distance from the light source. I was also unable to find any quantification of the 3D tracking volume, only of the overall recording arena, so I have no idea whether or not this question might be answerable from your data. Line 314 says that you did not test for range effects, which is not a very informative statement. I believe that you should test for range effects to the extent that your data allow, or explain why you cannot do so.

I am also unclear on what you're reporting about *Drosophila* species. Lines 255 & 290 reports them to be unaffected by light, but when you're discussing mixed Diptera results around line 165 no mention is made of the fruit fly results. This is a bit surprising, especially since fruit flies are much smaller than honey bees and likely much smaller than the mixed dipterans referred to around line 165, supposedly a section on small insects.

Regarding oleander hawkmoths – are you certain that this likely colony-bred animals have fully functional visual systems? I would appreciate a bit more discussion of why these animals did not respond in the usual way to the light stimulus; even if you don't have a certain answer it would be helpful to know if any possible explanations were eliminated.

I did not find the explanation of why a moon low on the horizon (line 302) does not produce a dorsal tilt and steering response very illuminating; as you're aware it is difficult to visually differentiate between a large distant light source and a small nearby source so the comment on line 306 that insects might ignore a distance source demands a bit more explanation.

Small comments:

Line 53 – what is a "short" wavelength in this context?

Line 72 – "across different orders of insects" how many orders?

Line 102 – missing "light" after "infrared"

Line 106 – what is the recording volume? Also, you're tuning for recording volume, not data volume.

Line 108 – could you describe the construction and interior of the tent more completely? In particular, is the tent fabric reflective at the wavelengths in question?

Line 123 – Why are only Common Darter dragonflies tested without the light source on?

Line 130 – "from-1" should be "from -1"

Line 199 – Here you describe 4 free parameters for the simulation; Figure 6 and the equation on line 505 mention only 3. Which is correct?

Line 400 "airspeed relative to the ground" – It is not clear to me what you mean with this phrase. Do you mean simply that wind speed and direction vary with time, or are you discussing the airspeed of an insect. Please rephrase.

Line 505 – Please number your equations! Also, Line 199 describes this model as having 4 free parameters

Line 696 – “it’s” should be “its”

Reviewer #3:

Remarks to the Author:

The manuscript “Why flying insects gather at artificial light” is well written, original and a timely contribution to our understanding of effects of artificial light at night on insects. I appreciate the approach of the authors to conduct field observation, a lab experiment and modelling to answer this question. Moreover, the authors explore the current hypotheses explaining attraction of insects to artificial light. They convincingly show that these hypotheses are not supported by the data and propose an alternative explanation, namely insects orient their dorsal axes towards light sources. However, the rejected hypotheses, such as lunar navigation, escape to light, attracted by thermal radiation and blinded by artificial light, mainly try to explain why so many insects die due to artificial light. This may be actually the ultimate question to be answered, rather than why there are so many insects around artificial light. When reading the manuscript, as answer why (some, ultimately not all?) insects die due to artificial light remains unclear. Although it is clear that insects respond by directing their dorsal axis towards the light source, it is not clear to me whether orbiting, stalling or inversion necessarily leads to collision with the lamp.

To better understand the generality of their findings, I would like to ask whether the authors can make a list of all (groups of) insect that were observed near an artificial light source. As several studies have indicated that larger insect species are more attracted to artificial light (and also the visibility in the experiment due to IR reflection), it would be informative to compare the insect species used for the experiment. Although I appreciate the captive flight experiments, the number of species (and individuals per species) is limited (and some did not show light-orienting behaviour). Moreover, I could not find sample sizes of the wild honeybees and mixed Diptera. Can the authors provide these numbers? Finally, I miss any quantification of the low light conditions. Can the authors give a bit more information on what low light conditions they recorded the flight of the insects? As the different light sources differ in spectral composition, this information on the light intensity is necessary, especially as insects may differ in their ability to perceive light of different wavelengths. It would be good to test the role of UV in the dorsal-light-response, especially for diurnal insects as they are thought to use UV light for orientation. Besides these points, the authors do a great job in explaining their methods well.

REVIEWER COMMENTS

Reviewer #1:

The authors tackle a question that humans have wondered about for thousands of years—why insects are attracted to light. Despite numerous hypotheses, there is surprisingly little data available in the literature. The authors tackle this with a combination of field and laboratory experiments and flight simulations. They present what appears to be an entirely novel hypothesis regarding why insects are attracted to light—corruption of a dorsal-light response for determining vertical orientation. This is similar, but crucially, quite distinct from the classic hypothesis that insects hold bright lights at a fixed orientation for nocturnal navigation.

Overall, the experiments are quite convincing of the proposed hypothesis and negate several other proposed hypotheses. The number of experiments conducted and the detail regarding the analyses conducted are quite impressive. In most cases, the results show very strong effects. I do have two small points that I would like to see addressed:

Major revisions

Comments: In some cases—specifically the first field experiments—there is insufficient data provided. For example, how often was each behaviour observed? Also, what were the insects that were observed? I realize it is not always possible to identify the species of insects, but the authors should do the best they can to identify at least the order.

Response: We have added a table listing the field videos collected with a qualitative survey of the different motifs observed (presence-absence, **Table 1**), the taxon present in the video (identified to different taxonomic levels, **Table 2**). It is worth noting that there is some bias in reporting because we were more likely to trigger recording events when the insect was in view for longer, and orbiting or stall behaviour caused the insect to stay in frame longer. Also, while trying to quantify motifs, in several videos multiple motifs were present at different parts of the flight, making it hard to quantify, and the final classification is somewhat subjective, but we provide the original videos in the final repository for future cross-verification.

Comments: In many cases, the authors present very convincing data, but fail to analyze it statistically (for example, the plots in Figure 5c). I have noted several places below where statistical analyses should be conducted in addition to the descriptive statistics presented in the current manuscript.

Response: Throughout the manuscript we have introduced further statistical testing to confirm our results. We have also applied a Bonferroni correction to our margin of significance to reduce type I errors. In particular, we have introduced Rayleigh Z-tests for the significance of the clustering of vector bearings (Lines 132-134). We have also added a statistical methods section at the end of the manuscript that describes how we prepared and tested our data (lines 595 - 616).

Overall, I consider this a ground-breaking piece of research—one of the most important papers I've seen published in this field in recent years. It has numerous implications for insect biology, conservation and behaviour and provides clear data to answer a question humans have pondered for millennia.

Response: We thank the reviewer for their feedback and appreciate their insight about the impact of the work.

Line-specific comments

Comments: Line 87 – Please provide some summary statistics regarding the duration of these recordings.

Response: We have added the mean and standard deviation of the recording durations to the text.

New Text: '(median duration 1.7 s IQR 1.9 s)'

Comments: Lines 86-99 It would be helpful to quantify the frequency of these behaviors. How often was each identified? Were there any trends in what insects exhibited which behaviors more frequently relative to insect order or body size? Also, what were the insects that you recorded? Can you at least provide a summary based on your ability to identify the insects to order or family level when possible?

Response: We have added summary tables (**Tables 1 & 2**) quantifying the frequency of response, as well as the identity of animals in the video and motif presence-absence in field videos. This information is given per-recording within **Supplementary Table 1**.

Comments: Lines 114-115 Again, what were the durations of these recordings?

Response While the per-recording durations are given within **Supplementary Table 3**, we have added the following summary statistics to the main text.

New text: (median track duration 1.7 s IQR 4.3 s).

Comments: Lines 120-123. The circular histogram plots look convincing, but you should also conduct statistics on the distributions, as currently in the figure all of the plots, including your control, have medians close to 90 degrees.

Response: We have confirmed the significance of vector clustering using Rayleigh z-tests. We have detailed our statistical methodology within a new section (Lines 594-615).

Comments: Lines 148-150. These values should be compared statistically.

Response: We have applied Wilcoxon rank-sum tests to compare the roll distributions and confirm their difference for each species. (Lines 165-166).

Comments: Lines 167-168. Are these insects active during the day, night or both?

Response: In the original manuscript, both the honey bees and mixed Diptera were primarily diurnally active. They were caught on the wing using a net during the day (14:00 - 17:00) and used shortly after capture. We have subsequently expanded this dataset to include nocturnally active small insects captured from a light-trap (23:00 - 01:00). This additional data is described within the updated manuscript, and we have also given a detailed break-down of the taxa used in a new table (**Table 4**). We have detailed the capture methods used for each taxon and use this as an approximation of their activity periods (lines 548 – 575). We assume that insects caught on the wing are primarily diurnally active, and insects arriving at the light-trap are primarily nocturnal/crepuscular.

Comments: Lines 182-183. Circular statistical tests should be conducted to demonstrate whether each distribution overlaps with zero.

Response: We have again used Rayleigh Z-Tests to confirm the significance of orthogonal velocity clustering.

Comments: Line 235. Here, you need to report the number of trials that were conducted and the number of times each moth switched orbiting direction after the light switch.

Response: We have conducted further trials including new taxa to this dataset. We have amended the text to clarify the observed frequency of light switching (lines 258 - 273). We have also added taxon-specific trial counts to the new **Table 5**.

Comments: Line 264-265. What are you basing this statement on? You didn't show any data of dorsal orientation of moths in the field, so is this just based on your qualitative review of the videos from the field?

Response: Yes, this statement is based on qualitative observations rather than dorsal axis reconstruction. We have amended the text of lines 293-295 to read: 'Qualitative observations from our videos in the field suggest that insects...'.

Comments: Line 326-328. What about the reflectance of the ground upon which lights are shining? Even if lights are pointed down, if the ground is well lit and reflective, would that cause insects to be attracted and make inversions such as the ones you demonstrated?

Response: This is a good point, and certainly the case when circumstances are correct, just as in our experiments with upward welling light. The best practices for reducing the harmful effects of artificial light, of which we mentioned just a couple, aim to minimise upward directed light, and ground reflection is one source of this. Particularly when lights are close to the ground and

bright, or when ground surfaces are especially reflective. We found this behaviour with a downward pointed tube about 1 ft away from a sheet, but not in other conditions. We now elaborate this point in the text to include ground reflectance.

Old text: “*Bright nearby lights can disrupt this mechanism and cause unintentional course alterations in insect flight. Taken together, reducing bright, unshielded, and upward facing lights will mitigate the impact on flying insects at night, when skylight cannot compete with artificial sources.*”

New text: “*Bright lights can disrupt this mechanism and unintentionally alter insect flight. Taken together, reducing unnecessary, unshielded, upward facing lights and ground reflections can mitigate the impact on flying insects at night, when skylight cannot compete with artificial sources.*”

Minor comments

Line 362 – Should also mention how the camera intrinsics were determined.

Response: We have amended our text to clarify we estimated both the intrinsic and extrinsic camera parameters using the inbuilt MATLAB camera calibration app. The new text reads as follows: ‘*We could then use the inbuilt MATLAB camera calibration app (Computer Vision Toolbox 10.3) to both detect the checkerboards in the views of both cameras and estimate both the intrinsic (optical centre, focal length, and radial distortion) and extrinsic camera parameters (relative camera orientation and translation).*’

Line 456 – Looks like a typo, check wording.

Response: We have corrected this typo.

Line 500 – citation here would be helpful.

Response: This statement refers to our observations of *S. striolatum* flying in total darkness. We have amended the text to clarify this point as such: ‘*We included this based on our observation that S. striolatum flying in total darkness still retains some degree of correct body attitude.*’

Lines 518-519 – How were these constants selected?

Response: We chose gains that reflect fast reactive feedback to control the measured error. To demonstrate that these gains are not ‘cherry-picked’ to provide the relevant behaviour, we ran the 300 random-gain simulations with a wide variation in all fitted parameters.

We have amended our in-text justification for the original gain set as follows: ‘*These parameters were chosen to reflect an insect flying at a relatively low Reynolds number (low terminal velocity), rapid aerial mobility (k values giving rapid reactions like those measured in insect pursuit flight controllers), and with lift and thrust profiles like those observed in our measured data (Supp. Fig. 2).*’

Comment: Line 535, dt should be replaced with Δt , since the change is not infinitesimal.

Response: We have amended the symbol to Δ rather than d.

Comment: Figure 6. Perhaps I missed this somewhere, but if you haven't already, please describe what the "sticks" of the "lollipops" represent in panels b-d.

Response: These 'sticks' represent the dorsal axes of the simulations at each drawn point. To highlight this, we have added the following text to the figure legend: '*We have represented the simulated position and dorsal axis of the agents by a point and vector line. Each point is colour coded with the associated speed.*'

Reviewer #2 (Remarks to the Author):

General Comments: I read your manuscript on your proposed explanation for why insects gather at artificial light and think that in general your work does a good job proposing and supporting a novel and interesting explanation for a widely observed phenomena of general interest. I do have some broader questions as well as some smaller notes that should be addressed.

Comments: One question that immediately came to mind was whether the size of your 3D tracking volume for the in-lab experiments permits any estimation of the "capture radius" of the lights you were using, or whether the data show any variation in general with distance from the light source. I was also unable to find any quantification of the 3D tracking volume, only of the overall recording arena, so I have no idea whether or not this question might be answerable from your data. Line 314 says that you did not test for range effects, which is not a very informative statement. I believe that you should test for range effects to the extent that your data allow, or explain why you cannot do so.

Response:

This is a salient and important further question about when insects are entrapped by artificial light. Our study set out to determine the most likely underlying behavioural mechanism that leads to light entrapment in flying insects. We thus targeted insects demonstrating entrapped flight-patterns around the light source. As a result, the majority of our data records the flight patterns of insects within 1 m of the light source and feature a great variety of insect taxa, not always identifiable in field recordings. In our section on long distance light attraction (line 345 - 363), we mention several research groups that employ techniques that track over much larger distances (but with less spatial and temporal precision). However, the data from these studies is still too sparse to reach any specific conclusions about more general boundaries of light-entrapment. The reviewer has picked up on a question of key interest to our research, and we are currently designing experiments to answer exactly this question, rather than attempt to reach an answer with our current data.

We have added the following text to our methods to clarify our recording volume in the field: '*In practice, this gave us a maximal recording volume of 1.5 m x 2 m x 1.5 m (Width x Depth x Height, with height aligning with gravity)*'.

We have also added the following text to clarify our recording volume in the high-resolution motion capture: '*The covered recording volume took the shape of a cylinder 1.6 m in diameter, 1.5 m tall, with the light-source at its centre. However, reconstruction depended on consistent marker visibility, which varied with distance from the light (and thus from the centre of the recording volume). As a result, contiguous stretches of reconstructed flight were generally within 0.5 m of the light source.*'

Comments: I am also unclear on what you're reporting about *Drosophila* species. Lines 255 & 290 reports them to be unaffected by light, but when you're discussing mixed Diptera results around line 165 no mention is made of the fruit fly results. This is a bit surprising, especially since fruit flies are much smaller than honey bees and likely much smaller than the mixed dipterans referred to around line 165, supposedly a section on small insects.

Response: We treated *Drosophila* separately in a section specifically devoted to the two species that did not show a strong behavioural response to light direction. We have amended the section dealing with small insects to reference that *Drosophila* did not show this response and indicated that they are discussed later in the text.

Comments: Regarding oleander hawk moths – are you certain that these colony-bred animals have fully functional visual systems? I would appreciate a bit more discussion of why these animals did not respond in the usual way to the light stimulus; even if you don't have a certain answer it would be helpful to know if any possible explanations were eliminated.

Response: We did not explicitly test that our research species had functional visual systems, but also did not have good reason to doubt they were able to see. Our captive *D. nerii* showed dark-adaptation in their eyes during night-time experiments, including a strong eye-shine reflection. However, as mentioned in the text, this species is entrapped around similar light-sources in the field (as recorded in: <https://doi.org/10.11609/jott.4694.11.5.13592-13604>). We have suggested that this may be the result of a suppression of the DLR in our captive *D. nerii* during experiments but cannot yet confidently say what causes this suppression. One potential answer may lie in the body posture and speed of flight within the enclosure. *D. nerii* within the tent flew slowly (median 0.9 m/s IQR 0.6 m/s), with a vertical body posture, similar to that seen during hovering. *Drosophila* adopt a similar vertical posture during flight. This is in strong contrast to the horizontal body attitude seen in groups such as dragonflies in flight. We aim in the future to look directly at flight differences between strongly light-affected species, and those seemingly immune, as this is potentially key to understanding more about the use of the DLR.

Comments: I did not find the explanation of why a moon low on the horizon (line 302) does not produce a dorsal tilt and steering response very illuminating; as you're aware it is difficult to

visually differentiate between a large distant light source and a small nearby source so the comment on line 306 that insects might ignore a distance source demands a bit more explanation.

Response: We added text to highlight that we are still fundamentally unsure about how insects account for the discontinuous brightness of the sky, and extremely bright small celestial sources like the sun and moon.

New text: *'Currently, we do not understand why insects do not dorsally tilt toward natural celestial sources of light. In some species, the DLR has two components, mediated separately by the insects' compound eyes and by the ocelli²⁸. Future work on the integration and luminance thresholds of these two components across multiple species would allow for a better understanding of how insects account for celestial light sources, and when artificial light destabilises them.'*

Small comments:

Line 53 – what is a “short” wavelength in this context?

Response: We have clarified we are referring to wavelengths in the Blue-UV range <450 nm.

Line 72 – “across different orders of insects” how many orders?

Response: We have amended the manuscript to read: “Across 10 different orders of insects”.

Line 102 – missing “light” after “infrared”

Response: Text amended to: *'Our motion capture used infrared light to track a custom marker frame'*

Line 106 – what is the recording volume? Also, you're tuning for recording volume, not data volume.

Response: We have amended the text to read 'recording volume'. We have also added a section to discuss the motion-capture recording volume to the methods: *'The covered recording volume stretched 0.8 m radially from the light source. However, reconstruction depended on consistent marker visibility, which varied with distance from the light (and thus from the centre of the recording volume). As a result, contiguous stretches of reconstructed flight were generally within 0.5 m of the light source.'*

Line 108 – could you describe the construction and interior of the tent more completely? In particular, is the tent fabric reflective at the wavelengths in question?

Response: We have added extra text to the methods to clarify the composition of the fabric tent walls (lines 469-470). We have also added a reflectance spectrum from the tent in

Supplementary Fig. 6.

New text: *'The tent was composed of white Joelastic fabric (J & C Joel) and the reflectance spectrum from the UV actinic tube light can be seen in **Supplementary Fig. 6.**'*

Line 123 – Why are only Common Darter dragonflies tested without the light source on?

Response: To clarify, for all species tested in the lab we compared control conditions (diffuse light similar to the sky) and treatment (a suspended point source). To ensure that the patterns under the point source conditions (circular flight) were not an artefact of the captive environment and the flight arena. We also tested a single species with the light off where we did not see orbiting flight, demonstrating our point. We used *S. striolatum* as it was the most abundant species and would most reliably fly within the laboratory arena.

Line 130 – “from-1” should be “from -1”

Response: We have corrected this typo.

Line 199 – Here you describe 4 free parameters for the simulation; Figure 6 and the equation on line 505 mention only 3. Which is correct?

Response: As on line 199 of the original manuscript, the model has 4 free parameters. 3 govern the dorsal tilting while the 4th is the agent terminal velocity (and thus simulated drag). This 4th parameter is included in equation 2.

Line 400 “airspeed relative to the ground” – It is not clear to me what you mean with this phrase. Do you mean simply that wind speed and direction vary with time, or are you discussing the airspeed of an insect. Please rephrase.

Response: We have changed the text to read ‘wind speed’ rather than airspeed.

Line 505 – Please number your equations! Also, Line 199 describes this model as having 4 free parameters

Response: We have revised the manuscript to have numbered equations.

Line 696 – “it’s” should be “its”

Response: We have corrected this typo.

Reviewer #3 (Remarks to the Author):

General comment: The manuscript “Why flying insects gather at artificial light” is well written, original and a timely contribution to our understanding of effects of artificial light at night on insects. I appreciate the approach of the authors to conduct field observation, a lab experiment and modelling to answer this question. Moreover, the authors explore the current hypotheses explaining attraction of insects to artificial light. They convincingly show that these hypotheses are not supported by the data and propose an alternative explanation, namely insects orient

their dorsal axes towards light sources. However, the rejected hypotheses, such as lunar navigation, escape to light, attracted by thermal radiation and blinded by artificial light, mainly try to explain why so many insects die due to artificial light. This may be actually the ultimate question to be answered, rather than why there are so many insects around artificial light. When reading the manuscript, as answer why (some, ultimately not all?) insects die due to artificial light remains unclear. Although it is clear that insects respond by directing their dorsal axis towards the light source, it is not clear to me whether orbiting, stalling or inversion necessarily leads to collision with the lamp.

Response: We thank the reviewer for their comments. We would like to clarify that the subject of this study was to identify the behavioural cause of the widely-observed light-entrapment phenomenon, rather than a more general investigation of the negative effects of artificial light on nocturnal insects. Light-entrapment is a core component of these effects of artificial light, but certainly not the only one. Artificial light at night can disrupt activity patterns, specific behaviours important for feeding & mating, and lead to higher predation risks. Other studies have detailed these effects, which can be either lethal or sub-lethal (Boyes et al. 2020, and Owens et al. 2020 for more in-depth accounts).

Comments: To better understand the generality of their findings, I would like to ask whether the authors can make a list of all (groups of) insect that were observed near an artificial light source. As several studies have indicated that larger insect species are more attracted to artificial light (and also the visibility in the experiment due to IR reflection), it would be informative to compare the insect species used for the experiment.

Response: We have added **table 2** detailing which insect orders were recorded at the light and showed the behaviour from the field videos collected. We have also characterised the spectrum of the lights used in **Supplementary Fig. 6**.

Comments: Although I appreciate the captive flight experiments, the number of species (and individuals per species) is limited (and some did not show light-orienting behaviour).

Response: Captive-flight experiments aimed to provide high resolution descriptions of the dorsal directions with the mo-cap system, while field experiments demonstrated the breadth of the response in natural environments (natural except for artificial light). With only field experiments we may not have observed animals unaffected by artificial light (as they might not be drawn in), but with only lab experiments, we could not have estimated the diversity of animals actually affected (as we hand select subjects). We now clarify the text to reflect that only a very small fraction of the animals did not show this behaviour in the lab. Further, to increase the sampling of species, we have added to our experiments with an additional round of fieldwork using a known set of 30 insect species (species details in the supplementary tables), spanning 6 orders with a single light condition.

Comments: Moreover, I could not find sample sizes of the wild honeybees and mixed Diptera. Can the authors provide these numbers?

Response: We have added further trials and taxa to our small insect assay. The text in this section has been updated accordingly. To clarify our sample sizes and taxa used, we have provided an additional table (**Table 4**) at the end of the manuscript.

Comments: Finally, I miss any quantification of the low light conditions. Can the authors give a bit more information on what low light conditions they recorded the flight of the insects? As the different light sources differ in spectral composition, this information on the light intensity is necessary, especially as insects may differ in their ability to perceive light of different wavelengths.

Response: Aside from our artificial light, conditions were profoundly dark, with the partial moon being the next most luminous source. Standard light metres are not sensitive enough to quantify this luminance, so we have used the method from Nilsson & Smolka 2021 (<https://doi.org/10.1098/rsif.2021.0184>), and found that without the artificial lights, insects flew below levels of $10\text{-}11.5 \log \text{ photons m}^{-2} \text{ s}^{-1} \text{ sr}^{-1} \text{ nm}^{-1}$, as in the new **Supplementary Fig. 7**. This method provides coarse spectral information as well. Insects approaching our lights are presumably flying as a result of both their regular circadian, time of day cues, and response to the artificial light. We also now include spectra of the lights in **Supplementary Figure 6**. We have added text in the manuscript to better explain this (Lines 615-628).

Comments: It would be good to test the role of UV in the dorsal-light-response, especially for diurnal insects as they are thought to use UV light for orientation.

Response: We agree with the reviewer and thank them for the suggestion. We tested both ultraviolet and visible white light in the field studies and found that visible light also provided similar results (orbiting, stalling, and inversions). However, we found a lower frequency of insects appearing at the light for white vs. UV. Due to time constraints, we chose to focus on the stronger and more frequent responses in the UV in order to obtain sufficient sample sizes. Hence, we did not report the data for the white LED. However, we have added sample videos of the flight motifs around white light (**Supplementary video 6**).

Comments: Besides these points, the authors do a great job in explaining their methods well.

Response: We thank the reviewer for their comments.

Reviewers' Comments:

Reviewer #1:

Remarks to the Author:

The authors have addressed all of my concerns and I believe the manuscript is ready for publication.

Reviewer #2:

Remarks to the Author:

Dear authors,

Thanks for your careful and thorough revisions in response to the referee comments on the first version of your manuscript. I have no further concerns and think this is a thorough and convincing demonstration of the source of artificial light attraction in insects flying at night.